

# An evolving Coupled Model Intercomparison Project phase 7 (CMIP7) and Fast Track in support of future climate assessment

John P. Dunne[1], Helene T. Hewitt[2], Julie Arblaster[3], Frédéric Bonou[4], Olivier Boucher[5], Tereza Cavazos[6], Paul J. Durack[7], Birgit Hassler[8], Martin Juckes[9], Tomoki Miyakawa[10], Matt Mizielinski[2], Vaishali Naik[1],
Zebedee Nicholls[11], Eleanor O'Rourke[12], Robert Pincus[13], Benjamin M. Sanderson[14], Isla R. Simpson[15], Karl E. Taylor[7]

[1]NOAA/Geophysical Fluid Dynamics Laboratory, Princeton, USA
[2] Met Office Hadley Centre, Exeter, UK
[3] School of Earth, Atmosphere and Environment, Monash University, Australia
[4] Laboratory of Physics and Applications (LPA), National University of Sciences, Technology, Engineering and Mathematics of Abomey (UNSTIM), Benin
[5] Institut Pierre-Simon Laplace, Sorbonne Université / CNRS, Paris, France
[6] Center for Scientific Research and Higher Education of Ensenada (CICESE), Baja California, Mexico.
[7] PCMDI, Lawrence Livermore National Laboratory, Livermore, CA, USA
[8]Deutsches Zentrum für Luft- und Raumfahrt (DLR), Institut für Physik der Atmosphäre, Oberpfaffenhofen, Germany
[9] University of Oxford, and UKRI STFC, UK
[10] Atmosphere and Ocean Research Institute, The University of Tokyo, Kashiwa, Japan
[11] Climate Resource, Berlin, Germany;Energy, Climate and Environment Program, International Institute for Applied Systems Analysis (IIASA), 2361 Laxenburg, Austria; School of Geography, Earth and Atmospheric Sciences, The University of
Melbourne, Melbourne, Victoria, Australia
[12] CMIP International Project Office, ECSAT, Harwell Science & Innovation Campus, UK
[13]Lamont-Doherty Earth Observatory, Columbia University, Palisades NY USA
[14] CICERO, Oslo, Norway
[15] NSF National Center for Atmospheric Research, Boulder, Colorado, USA

*Correspondence to*: John P. Dunne (john.dunne@noaa.gov)

**Abstract.** The vision for the Coupled Model Intercomparison Project (CMIP) is to coordinate community based efforts to answer key and timely climate science questions and facilitate delivery of relevant multi-model simulations through shared infrastructure for the benefit of the physical understanding, vulnerability, impacts and adaptations analysis, national and international climate assessments, and society at large. From its origins as a punctuated phasing of climate model
intercomparison and evaluation, CMIP is now evolving through coordinated and federated planning into a more continuous climate modelling program. The activity is supported by the design of experimental protocols, an infrastructure that supports data publication and access, and the phased delivery or "fast track" of climate information for national and international climate assessments informing decision making. Key to these CMIP7 efforts are: an expansion of the Diagnostic, Evaluation and Characterization of Klima (DECK) to include historical, effective radiative forcing, and focus on $CO_2$-emissions-driven
experiments; sustained support for community MIPs; periodic updating of historical forcings and diagnostics requests; and a collection of experiments drawn from community MIPs to support research towards the 7th Intergovernmental Panel on





Climate Change Assessment Reporting cycle, or "AR7 Fast Track", and climate services goals across prediction and projection, characterization, attribution and process understanding.

## 1 Introduction

The Coupled Model Intercomparison Project (CMIP) is an international research activity that develops coordinated experimental protocols within the World Climate Research Programme (WCRP) for global atmosphere-ocean-land-ice coupled climate and Earth System Models (ESMs) and facilitates the distribution, interpretation, and use of simulation output. ESMs represent the time evolution of the climate and the statistical characteristics of the weather through a combination of the representation of the dynamical equations of motion and equations describing the physics and thermodynamics of the

interactions between radiation, clouds, aerosols and the coupled hydrosphere, geosphere, biosphere, and cryosphere. Preceding phases of CMIP (Meehl et al., 1995; 2000; 2007;Taylor et al., 2012; Eyring et al., 2016) have made evident how the evolution of ESMs has improved the representation of the Earth system through increases in spatial resolution (initially tens of degrees to now around a quarter of a degree), comprehensiveness (including carbon cycle, atmospheric chemistry, aerosols, biogeochemistry, ecosystems, ocean acidification, sea level rise, and human drivers), and granularity (ensembles of models

assessing structural uncertainty, detection and attribution, predictability, sensitivity to feedbacks, statistics of extremes, etc). There are, however, persistent model structural uncertainties and biases through the generations of CMIP that continue to require model development and assessment to ensure that these models are able to produce the most accurate predictions for the climate system moving forward.

As self-consistent representation of physics, biology, and chemistry on weather to climate time scales, each ESM contributing

to past phases of CMIP has represented one combination of choices along the many dimensions of the multiverse of models (Figure 1). In particular, in addition to representing water and energy cycles and associated dynamics as in physical climate models, ESMs broaden the focus to questions in which the coupling between chemistry and/or the carbon cycle and the physical climate system plays a key role, for example exploring interactions between anthropogenic emissions and climate as mediated by biogeochemical cycles (Sanderson et al., 2024).

As an international research activity within WCRP, CMIP supports the WCRP science priorities of "Fundamental understanding of the climate system", "Prediction of near term evolution of the climate system", "Long term response of the climate system", and "Bridging climate science and society." As described in Meehl (2023) and Stevens (2024), the origin of CMIP was to systematically assess coupled models – to characterize their biases, interactions, and response – and evaluate the effectiveness of ongoing model development efforts to address structural model issues or incorporate new processes.   In early

CMIP phases, the core CMIP experiments were those commonly performed by individual modeling centers during their model development cycles.   A key to the scientific value of the model intercomparison research was that all models were run under the same experiment conditions.   In particular, the same forcings were imposed on the models.   With CMIP's early successes, the forcings were improved and extended for each of its successive phases.  Equally important to CMIP's research appeal and



impact were the strict standards imposed on the data produced by the models and the multi-model archive of all CMIP data,
supported by a specially purposed software infrastructure (Durack et al., 2025). With all models providing publicly available results in the same format and structure, the same downloading tools and analysis code could be applied to all models without altering how the model output was ingested.

As a publicly available ensemble including state-of-the-art coupled model contributions from centers around the globe, CMIP collects simulations of varying levels of structural idealization from many physical climate models and ESMs. This
international effort supports a wide range of science activities by providing a combination of idealized and single forcing experiments for the scientific community to interrogate and build a robust scientific literature underpinning mechanistic and process understanding of the complexities of climate change in the Earth system. More realistic historical and projection simulations also support investigation into quantification of change and application to a broad range of societally relevant impacts.

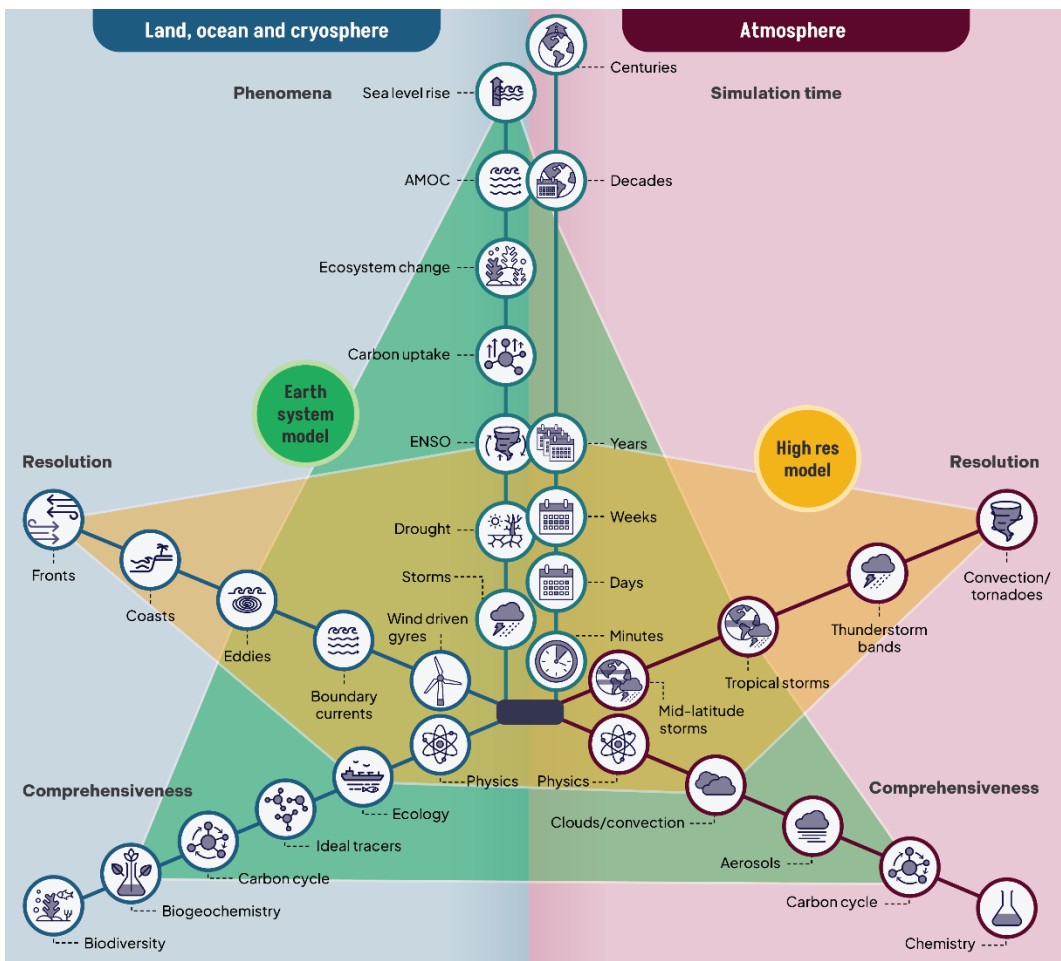


**Figure 1: Earth system modelling as part of the multiverse of modeling approaches across resolution, comprehensiveness and simulation time. Atmospheric aspects are shown in red and ocean aspects in blue. Note that ensemble size, experiments/scenarios, precision, accuracy, availability and familiarity also come into play in the search for efficiency and robustness.**



Beyond uncovering the systematic behavior of coupled models and the associated uncertainty in climate behavior and the
underlying response, CMIP simulations have also proven useful across the scientific community for exploring ideas without
having to design unique experiments and run simulations in house wherein the underlying physics is elucidated through
intercomparison. Examples are wide ranging including tropical (Bellenger et al., 2014; Planton et al., 2021) and extra-tropical
variability (Simpson and Polvani, 2016; Zappa and Sheppard, 2017), the behavior of temperature and precipitation extremes
(Seneviratne and Hauser 2020; Borodina et al., 2017), understanding the factors that influence modelled climate sensitivity
(e.g., Zelinka et al., 2019), and the connections between the representation of present-day climatology or processes and future
projected change (e.g., Hall et al., 2019).

In addition to the systematic characterization of climate mechanisms, CMIP has supported national and international
assessments in the provision of climate response to first idealized forcing followed by selected community-developed scenarios
of projected forcing (Meehl et al., 2007). Projected climate change in coupled models due to increased greenhouse gas forcing
has also been part of every Intergovernmental Panel on Climate Change (IPCC) report since its inception. The first climate
change projections made with climate models used instantaneously doubled $CO_2$ concentrations to estimate what has become
known as Equilibrium Climate Sensitivity (ECS; Manabe and Stouffer, 1980). A second idealized sensitivity experiment
incorporated transient $CO_2$ forcing — increasing $CO_2$ 1% per year to assess the Transient Climate Response (TCR; Mitchell
et al., 1990) — followed as part of CMIP2 (Meehl et al., 1997) in support of the IPCC third (Cubasch et al., 2001) Working
Group I assessments, respectively. Idealized simulations were complemented by sets of more realistic historical and projected
scenarios in subsequent iterations of the protocol. One of the key roles of CMIP has been to provide one line of evidence on
the likely range of $CO_2$ climate sensitivity in IPCC Assessments. The role for CMIP has broadened to general use for
systematically sampling and characterizing model diversity as an element of uncertainty in a range of climate applications.
These include not only response to $CO_2$ but also aerosols, ozone, carbon system and others, and evolving from an initial focus
on the response in temperature and precipitation to the response in extremes, such as drought, heat waves, monsoons and
tropical storm statistics, and other climate indicators such as sea ice, ocean circulation and sea level rise, with implications
across economic and societal sectors for agriculture, energy, transportation, infrastructure, and resilience among many others.
Together, these activities support assessment and other climate services with increased understanding and projections across
a suite of potential futures in support of climate resilience, adaptation and mitigation planning, policy analysis, and decision-
making. Beyond direct contribution to national and international climate assessments, CMIP increasingly also supports climate
service activities including downscaling through international WCRP projects such as the COordinated Regional Downscaling
EXperiment (CORDEX; https://cordex.org/; Giorgi and Gutowski, 2015; Gutowski et al., 2016) and the Regional Information
for Society (RIfS; https://www.wcrp-rifs.org/), Inter-Sectoral Impact Model Intercomparison Project (ISIMIP; Warszawski et
al., 2013), Sea Level projections via FACTS (Kopp et al., 2023), Vulnerability, Impacts, Adaptation, and Climate Services
(VIACS; Ruane et al., 2016) and  government services such as the Copernicus Interactive Climate Atlas
(https://atlas.climate.copernicus.eu/atlas) which resulted from previous atlases of the IPCC Reports (e.g., Gutierrez et al.,
2021). CMIP has been also applied to non-governmental, non-profit climate change attribution evaluation reports and real time





diagnostics of high impact extreme events around the world such as World Weather Attribution (https://www.worldweatherattribution.org/; Otto, 2017, 2023) and Climate Central (https://www.climatecentral.org/; Gilford

et al., 2022). CMIP results have been incorporated in climate vulnerability and readiness analyses for governmental policy, insurance, military preparedness, Non-Governmental Organizations, media communication, and commercial sector use, among others. CMIP thus serves at least three roles: a focal point for scientific inquiry across a range of idealizations; a source of information for the policy-relevant assessment of current understanding; and as a plausible representation of possible futures used both as a direct source of information or indirectly as a source of inputs through additional bias correction, sub-selection,

climate attribution, downscaling or impacts modelling for climate services. Balancing the needs of research, assessment, and applications has historically been one of the challenges of designing CMIP phases because the burdens fall entirely on the research community. Though assessment and service needs might well be better met by a more sustained application of ESMs to routinely updated forcings (Schmidt et al., 2023; Jakob et al. 2023; Stevens 2024), the infrastructure for such a sustained approach is not yet in place at either national or international levels. In the absence of non-research infrastructure for climate

and Earth system modelling, the present experimental design includes some components that might fruitfully be taken up outside the research community, but a set of immediate service needs remain an ongoing component of the project.

The design of CMIP7 responds to the experiences obtained during CMIP6 and subsequent surveys and community feedback. Changes to the protocol and organization, described more fully below, are intended to address community concerns by reducing the burdens of simulation and data provisioning for contributors, facilitating more nimble community-driven efforts, and more

clearly distinguishing among those aspects supporting science, assessment, and service. The goals of CMIP7 are thus to provide a framework supporting: 1) the rich diversity of small-scale research built in CMIP6, 2) continue to enable episodic and punctuated participation and intercomparison and 3) facilitate more sustained participation with continuous and responsive support.

Though simulations made for previous phases of CMIP have been a rich resource for developing understanding, it is worth

asking whether the research community stands to benefit from another iteration of the project. The design presented here seeks to emphasize the value obtained from an updated set of simulations by Earth system models. That value arises from three main developments. First is the accumulation of a longer, richer observational record encompassing a wider range of conditions and the accelerating emergence of the climate change signal from climate variability. Second is the ongoing development and increasing comprehensiveness of ESMs aided by observational advances including increasingly diverse satellite observations

of atmospheric composition, land characteristics, and ocean ecology. These models need to be evaluated, and their behavior understood to interpret the results in the context of these new constraints. Third is the formulation of new questions about the co-evolution of natural and human systems, especially as related to the trajectory of the carbon cycle and its response to human activities, and the elaboration of models designed to address them. The design of CMIP7 is focused on four new research questions described in the next section for which understanding is evolving rapidly and new simulations promise to provide

sharper insight. This section is followed by the CMIP7 guidance on protocols for the Diagnostics, Evaluation, and Characterization of Klima (DECK) and "AR7 Fast Track" experiments and their context in the evolving role of CMIP.





## 2 Guiding Research Questions

The scientific component of CMIP7 focuses on four guiding research questions for which moderately sized ensembles of ESM simulations hold promise for substantial progress. These questions are more focused on ESMs — and hence more timely and ephemeral — than those posed for CMIP6. A key opportunity permeating all these questions is the ability to confront the modelled representation of historical trends with a further eight years or more of the observational record past the 2014 termination from CMIP6 and new data constraints including the Earth radiative imbalance (Schmidt et al., 2023).

### 2.1 Patterns of sea surface change: How will tropical ocean temperature patterns co-evolve with those at higher latitudes?

*Description:* The spatial pattern of sea surface temperature across the vast tropical Pacific has global implications through teleconnections and radiative feedbacks (e.g., Kang et al., 2020). Models in earlier generations of CMIP consistently predicted that the global warming Sea Surface Temperature (SST) signal in the tropical Pacific would resemble El Niño with an enhanced warming in the eastern equatorial Pacific (e.g., Cai et al., 2014; Wang et al., 2017). However, the AR6 report states that "there is no CMIP6 model consensus for a systematic change in intensity of ENSO SST variability over the 21st century," (Cai et al., 2022). Moreover, over the last several decades a signal of enhanced warming in the western Pacific and slight cooling in the eastern Pacific has emerged i.e., the opposite from that predicted by models on average (Coats and Karnaukas 2017; Seager et al., 2019). At the same time, a cooling has occurred in the Southern Ocean in the observational record in contrast to the expected warming based on CMIP simulations and there is growing evidence of a connection between trends in the Southern Ocean and those in the tropical Pacific (Dong et al., 2022; Kang et al., 2023). It is becoming increasingly clear that the SST trends observed in both the tropical Pacific and the Southern Ocean are at the very edge or outside the range of those simulated by CMIP6 models (Wills et al. 2022, Seager et al., 2022) raising concerns that models are able to capture neither the externally forced trend nor the magnitude of internal variability (or both) in these regions (Watanabe et al., 2024). The research community is now working to understand the origins of this discrepancy and there are now indications that unresolved processes such as ocean eddies (Yeager et al., 2023), melt water forcing (Dong et al. 2022, Schmidt et al, 2023), or recalcitrant biases, such as the double Intertropical Convergence Zone (Watanabe et al., 2024) or the cold bias in the equatorial cold tongue (Seager et al. 2019), may be playing a role. Transient or permanent shifts in SST patterns may also drive changes in the strength of some feedbacks (especially those mediated by clouds) and decadal changes in the Pacific (e.g., Li et al., 2023) with an impact on our understanding of the ongoing and future climate response with implications for climate sensitivity and time to 2C warming (Armour et al., 2024). Related to this key concern is the need for better joint understanding of historical and recent aerosol forcing and warming trends which appear to rule out high warming models in CMIP6, suggesting that the mechanisms behind both Earth's radiative balance and temperature changes may require a reassessment. In contrast to long-term trends, recent observational trends of the ocean heat content (OHC) of the upper 2000 m during 2005-2020 show significant warming in the tropical Pacific, subtropical oceans and the Southern Ocean, which reflect the El Niño-like structure and recent Pacific decadal shifts (Li et al., 2023). This study also documents a strong acceleration in global ocean warming since the 1990s,



amounting to >25% increase in OHC during 2010–2020 relative to 2000–2010, and nearly a twofold increase during 2010–2020 relative to 1990–2000. This accelerated warming can have important implications for future SST trends and climate change.

*Why expect progress now?* Research through CMIP7 on the sea surface warming patterns bolstered by a combination of advances including improved process understanding from the Tropics community (e.g., Ray et al., 2018; Planton et al., 2021),

longer observational time series of historical forcings, improved forcings constrained by new satellite and in situ observations, better understanding of forcing uncertainty and internal variability, novel ideas about teleconnection mechanisms, potential reductions in biases in the double ITCZ, Walker circulation and ENSO through model improvements and increased resolution in the atmosphere and ocean (e.g. Yeager et al. 2023), may all help. Particular emphasis will be on the combination of improved and longer historical large ensembles in the context of the Detection and Attribution Model Intercomparison Project (DAMIP)

and Aerosols and Chemistry Model Intercomparison (AerChemMIP) to further untangle the role of regional aerosol forcings.

## 2.2 Changing weather: How will dangerous weather patterns evolve?

*Description:* Large scale patterns of climate play a critical role in maintaining background conditions that can trigger many weather extremes including hurricanes and other storms, storm surges and tornadoes, floods, droughts, heat waves, wind droughts, and monsoons whose frequency and/or intensity have started to change. Understanding how these large-scale

patterns will further respond to climate change is key to providing regional decisions with actionable information on climate change adaptation. The CMIP6 large ensembles were of incredible value in highlighting the role of internal climate variability and in quantifying the level of discrepancy between model behavior and the historical record (e.g., Wills et al, 2022). The more active hydrological cycle projected under warming is expected to increase the potential for large storms consistent with several recent examples of record-breaking storms such as the upper-tropospheric cut-off lows (known as DANA in Spanish) that

produced severe floods in Valencia and other regions of Spain in November 2024, and rapid intensifying hurricanes, such as Otis in 2023 in the Eastern Tropical Pacific (Garcia-Franco et al., 2024) and Helene and Milton in the southeastern United States (Clarke et al., 2024), and their increase attributed to climate change (Bhatia et al., 2021; Clarke et al., 2024). There is a growing need to know how to adapt to rapid and unexpected changes, which require more robust and finer resolution projections, and better understanding of the causes and shifts in spatial and temporal distributions of dangerous and impactful

weather patterns for this information to be actionable. Given that many extreme events are threshold behavior based (e.g. tropical cyclones, ice melt, coral bleaching, etc.), these priorities motivate improvements in the mean state of climate models to better match absolute historical temperatures as well as the change.

*Why expect progress now?* Better statistics of rare events and extremes remain critical to meet the enormous research and

societal challenges at hand. One key role of CMIP in the multiverse of modelling efforts is the running of multi-centennial coordinated simulations supporting characterization of frequency distributions of infrequent events. The CMIP7 focus on $CO_2$-emissions-forced models will allow for novel investigation of extremes under climate stabilization. Though the CMIP7





protocol does not specify large ensembles, some modelling centers may contribute large ensembles (as in CMIP6) allowing for better characterization of rare events. The considerable effort devoted to understanding the causes of the high ECS obtained

in many models in CMIP6 may lead to improved representation of historical climate change (Meehl et al., 2020; Golaz et al., 2022). Finally, given anticipated modest enhancements in resolution for some models and how similar models behave across a range of resolutions (Roberts et al. (2024)), CMIP7 should include improved projections of extremes such as hurricane frequency. While full participation with km-scale ultra-high resolution simulations in which convection may be explicitly represented, known as convection-permitting (e.g., Coppola et al., 2020; da Rocha et al., 2024) remains in the future, CMIP7

simulations will also be complemented by regional downscaling efforts such as CORDEX.

### 2.3 Water-carbon-climate nexus: How will Earth respond to human efforts to manage the carbon cycle?

*Description:* State of the art coupled carbon cycle climate modelling sits at the intersection of climate, ecosystems, hydrology, biogeochemistry and societal modelling, with the future resilience of natural and potentially human-modulated carbon sinks being the key uncertainty in relation to climate stabilization and warming reversal. Quantification of the land and ocean

processes responsible for the historical carbon concentration response to anthropogenic $CO_2$ emissions constitutes an important step forward in demonstrating model robustness. Critical to understanding the future carbon budget is quantifying how vegetation responds to changing climate and how soils respond to warming, moisture, and thawing in the context of a changing microbial communities (e.g., Chase et al., 2021) and how the processes that determine vegetation growth interact with soil microbial functioning and will respond to changing climate (Lennon et al., 2024). Beyond this need for better historical and

future natural system understanding, exploration of the many dimensions of Carbon Dioxide Removal (CDR) is critical to understand the vulnerabilities of ecosystems to natural and anthropogenic drivers such as climate variability, ecosystem management, land use fires, and pests. While previous carbon mitigation scenarios have placed a large reliance on the viability of BioEnergy with Carbon Capture and Storage (BECCS; IPCC Special Report on Land, 2018), there remain deep and multidimensional uncertainties such as competition for water and land use between BECCS, afforestation, biodiversity

protection and agriculture. Constraining land carbon uptake depends on knowledge of the ocean carbon uptake, but with the discrepancy between current surface estimates based on $pCO_2$ observations and prognostic biogeochemical models (RECCAP2; Friedlingstein et al., 2023) recently increasing to 1 Pg/yr, our ability to confirm the effectiveness of prospective land or ocean CDR is limited. Ocean CDR effectiveness, durability, vulnerability and overall additionality of proposed solutions such as iron fertilization, alkalinization, $CO_2$ injection, and carbon capture in seaweed has only recently been

explored. Also uncertain is how ocean acidification will evolve under continued stratification and will affect oceanic ecosystems in the context of CDR.

*Why expect progress now?* Building on the introduction of Coupled Carbon-Climate ESMs in CMIP5 with more experiments added in CMIP6 towards process understanding, CMIP7 shifts the scientific focus to their response to $CO_2$ emissions and removals and the coupled mechanisms necessary to achieve climate stabilization. As such, CMIP7 is expected to include more

comprehensive process representation of coupled carbon-climate in ESMs including the non-linear role of biogeography, land



use, fires, permafrost and microbes. New experiments forced by $CO_2$ emissions (Sanderson et al., 2024) evaluate the robustness of the Transient Climate Response to cumulative Emissions (TCRE) under net zero and net negative global emissions. Improved ESMs in CMIP7 will be better positioned to contextualize the assumptions and uncertainties associated with carbon cycle response and removals used to deliver climate forcings from Integrated Assessment Models, and characterize climate response and feedbacks.

## 2.4 Points of no return/ratcheting: What are the risks of triggering irreversible changes across possible climate trajectories?

*Description*: In AR6, the IPCC defined a Tipping Point as "A critical threshold beyond which a system reorganizes, often abruptly and/or irreversibly" and highlighted several possible tipping elements including Atlantic Meridional Overturning Circulation (AMOC) collapse, Amazon die-back, poleward migration of temperate forests, Sahel greening, sea level rise/ice sheet collapse, and Arctic warming and associated loss of permafrost and carbon release (Lee et al., 2021). For example, as projected forest dieback and demography shifts largely depend on the potential for drought and both thermal and hydrological factors (Drijfhout et al., 2015), representation of climate vegetation interactions is key to robust characterization of potential change. In the case of the Amazon, for example, recent work focused on observations suggests that resilience declines have already begun which could set the stage for major change (Boulton et al., 2022) while modeling suggests that increase in fire over this century (Allen et al., 2024). In the case of potential Southern Ocean changes and Antarctic ice sheet collapse, the state of uncertainty remains extremely high with CMIP6 era models lacking fidelity in key processes such as representation of the Antarctic slope current and land-ice interactions or agreement in change (Fox-Kemper et al., 2021). Proposed mechanisms of irreversible and potential sudden change are manifold (Lenton et al., 2008; Drijfhout et al., 2015), and scientists and society alike are interested in identifying early signs of tipping points and designing early warning systems as an adaptation to climate warming, particularly as these tipping points influence climate impacts.

*Why expect progress now?* Analysis of CMIP6 models has identified emerging advances of tipping processes such as fire (e.g., Allen et al., 2024) while application of Machine Learning (ML) methods has brought new insight into early detection of tipping points (e.g., Bury, et al., 2021). Ongoing improvements in historical simulations of warming and more constrained ECS will give greater confidence in results while the inclusion of more advanced ESMs forced by emissions combined with the provision of overshoot scenarios will provide the opportunity to explore the possibility of irreversible changes even with climate stabilization. Recent results from paleoclimate studies such as exploration of the Green Sahara during the mid-holocene (Hopcroft and Valdes, 2021) also provide the opportunity to confront climate models with possible process-driven storylines of how tipping points may occur. New capabilities in CMIP7 models including coupled ice sheet models, expanded biogeochemical processes (including dynamic land use type) and higher resolution models will enable new insights on tipping points.



## 3. CMIP7 Experimental Design: Expanded Baseline Experiments and the AR7 Fast Track

The CMIP6 experiment design (Eyring et al., 2016) decentralized scientific leadership through a new process of endorsing MIPs while retaining responsibility for defining a small number of simulations to characterize the basic behavior of each

participating model through the DECK and historical experiments. This led to a successful expansion of CMIP into new areas of science and new communities, including specific requests from a wide range of groups working on climate impacts (e.g., through VIACS, Ruane et al., 2016). Despite efforts to harmonize requests for experiments and data across MIPs, however, this rapid expansion also led to significant burdens on participating modelling centers. Efforts to present the requirements of the new MIPs in a consolidated form led to a perception of a monolithic request. This pressure of requests coming from many

independent MIPs was exacerbated by the perceived need to produce all simulations early enough to be included in the IPCC's Sixth Assessment Report (AR6) – conflating research, assessment, and service timelines. These and other issues highlighted in feedback from the modelling community, however, including responses to a CMIP6 community survey (https://zenodo.org/records/11654909) similar to the one after CMIP5 (Stouffer et al., 2017), motivated an approach of simultaneously less centralized coordination but more targeted recommendations on those CMIP7 experiments most likely to

support the climate service and process understanding needs of the IPCC assessment versus the more general application of models in community MIPs.

The CMIP7 protocol responds to these survey results by more clearly distinguishing among simulations intended to: 1) systematically characterize model behavior, 2) establish ranges for future climate change under different emissions trajectories, and 3) target high priority scientific questions (Section 2) and 4) maintain explicit alignment with the IPCC assessment process.

To this end, the DECK is modestly expanded, community-driven and scientifically motivated MIPs are supported more broadly but encouraged to run on self-determined timelines, and assessment reports are supported by identifying and prioritizing small thematic sets of simulations, drawn from the MIPs, of particular relevance to informing such reports (Figure 2). This section includes a description of the first such set, a "fast track" focused on the four motivating questions on a timeline allowing inclusion in the upcoming IPCC Seventh Assessment Report (IPCC-AR7). The design incorporates extensive community input

and seeks to energize research inspired by emergent advances and modelling center priorities rather than impose a single monolithic view from any single organizational perspective or stakeholder demand.

Acknowledging that details of the protocols described here are subject to modest change over time, the current (and all previous) versions, and the differences between them, will be made available as living documents through the CMIP website (https://wcrp-cmip.org/).



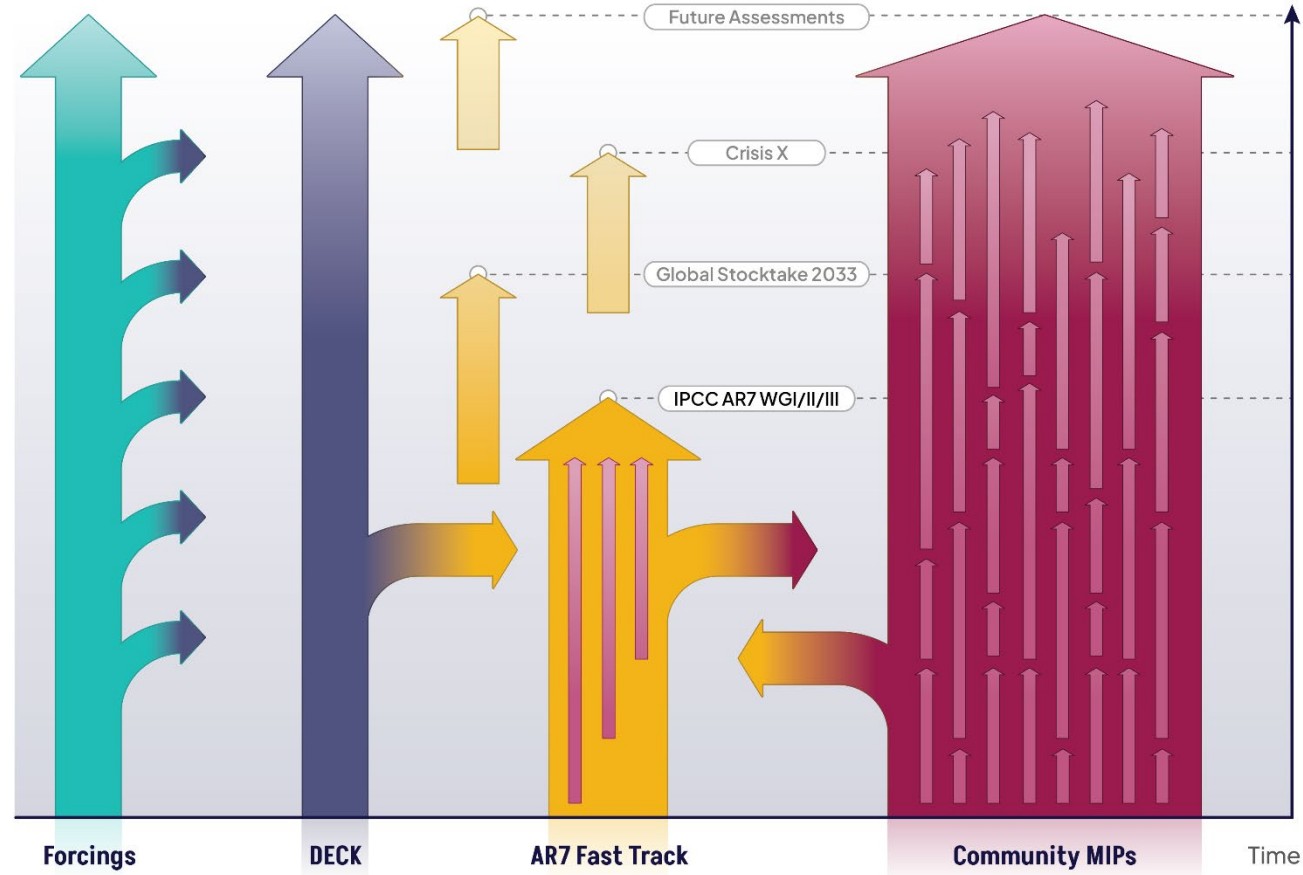

**Figure 2: Schematic of the evolving CMIP design into an even more continuous approach with a continued DECK, regular updates and extensions of forcings, targeted "Fast Track" experiment sets starting with the "AR7 Fast Track", and CMIP infrastructure, standards, and tools also supporting ongoing science activities through community MIPs.**

**3.1 Diagnosis, Evaluation and Characterization of Klima (DECK) Experiments**

CMIP6 introduced a set of mandatory baseline experiments aimed at the Diagnosis, Evaluation and Characterization of Klima (German for Climate), all of which were performed for CMIP5 and most in prior iterations of CMIP (Eyring et al. 2016) and serve as the CMIP "entry card" for participation. The CMIP7 DECK is based on the same experiments (Table 1, short names in italics) but is expanded modestly be adding a) the historical simulation, b) a small set of "fixed-SST" experiments to characterize effective radiative forcing, and c) an expanded protocol to facilitate participation with ESMs capable of running with interactive $CO_2$ forced by emissions.

The expanded DECK is intended to allow for more complete description and characterization. Historical simulations (*historical* or *esm-hist*), which are most often interpreted in the context of more idealized experiments, are included in the DECK because they are key for characterizing model behavior over the observed historical record. Protocols remain formally unchanged from CMIP6 although more detailed guidance for models simulating biogeochemical mechanisms (and thus



concentrations of $CO_2$ given emissions) and specifications of forcings (Table 1) is provided below. Because physical and compositional perturbations, whether specified as a forcing or computed internally, do not fully specify radiative perturbations driving climate change (e.g., Soden et al. 2018; Smith et al. 2020), the CMIP7 protocol modestly expands the DECK to characterize model-specific effective radiative forcing (as was "strongly encouraged" in CMIP6). Three atmosphere-only experiments with fixed model-specific pre-industrial sea surface temperature and sea ice concentration fields are added to the

DECK following protocols developed for CMIP6 by the Radiative forcing Model Intercomparison Project (Pincus et al. 2016; Table 1).   The abrupt 4xCO2 experiment is further modified with a recommendation to extend the simulation out to 300 years, if possible, to provide a more robust estimate of the Equilibrium Climate Sensitivity (Rugenstein et al., 2020; Dunne et al., 2020).

**Table 1: Overview of the CMIP7 DECK with experiment short names, brief experiment descriptions, the forcing methods, as well as the start and end year and minimum number of years per experiment, and its main purpose. The DECK is used to characterize the CMIP model ensemble. Any size of ensemble is acceptable but the protocol requests submissions of least three ensemble members for the CMIP historical simulation as requested in DAMIP. Large ensembles of the Atmospheric Model Intercomparison Project (AMIP) simulations forced by SST and Sea Ice Concentrations (SIC) are also encouraged. In the "forcing methods" column, "All"**

**means all natural and anthropogenic forcings including greenhouse gases, aerosols, and land use as described in Table 2. Experiments start on 1 January and end on 31 December of the specified years. The recommended piControl minimum experiment length is defined below; however, to ensure broad simulation data use, piControl temporal coverage should extend across the equivalent period (after initialization) to that in the full historical and future scenario (with extension) periods. The plus (+) sign indicates that beyond meeting the basic DECK requirements, the total number of simulated years would depend on the number of**

**ensemble members, whether the piControl will follow the Fast Track guidance of 150 year abrupt-4xCO2 extension to 300 years and whether the scenarios and their extensions are being run.  Further information of anthropogenic forcing for CO2 emission- and concentration- forcing is provided in Section 3.1.1.**

| Experiment short name | Experiment description | Anthropogenic Forcing | Volcanic Forcing | Solar Forcing | Start Year | End Year | Main purpose |
|---|---|---|---|---|---|---|---|
| *amip* | Observed SSTs and SICs prescribed | Time-varying | Time-varying | Time-varying | 1979 | 2021 | Evaluation, SST/sea ice forced variability |
| *piControl and/or esm-piControl* | Coupled atmosphere-ocean pre-industrial control | All 1850, $CO_2$ prescribed concentration or emission | Fixed mean radiative forcing matching historical simulation (i.e. 1850–2021 mean) | Fixed mean value matching first two solar cycles of the historical simulation | 1 | 400+ | Evaluation, unforced variability |



| | | | | (i.e. 1850–1873 mean) | | | |
|---|---|---|---|---|---|---|---|
| *abrupt-4xCO2* | $CO_2$ prescribed to 4 times preindustrial | Same as piControl except $CO_2$ concentration prescribed to 4 times piControl | Same as piControl | Same as piControl | 1 (branching from year 101 or later of piControl) | 150+ (300) | Equilibrium climate sensitivity, feedback, fast responses |
| *1pctCO2* | $CO_2$ prescribed to increase at 1% yr-1 | Same as piControl except $CO_2$ prescribed to increase at 1% yr-1 | Time varying | Time varying | 1 (branching from year 101 or later of piControl) | 150 | Transient climate sensitivity |
| *historical* and/or *esm-hist* | Simulation of the recent past | All time varying, $CO_2$ prescribed concentration or emission | Same as piControl | Same as piControl | 1850 | 2021 | Evaluation |
| *piClim-Control (amip)* | Preindustrial conditions including SST and SIC prescribed | All 1850, $CO_2$ prescribed concentration | Same as piControl | Same as piControl | 1 | 30 | Baseline for model-specific effective radiative forcing (ERF) calculations |
| *piClim-anthro (amip)* | As *piClim-Control* except present-day anthropogenic forcing | All 2021, $CO_2$ prescribed concentration | Same as piControl | Same as piControl | 1 | 30 | Quantify present-day total anthropogenic ERF |





| | | | | | | | |
|---|---|---|---|---|---|---|---|
| *piClim-4xCO2 (amip)* | As *piClim-Control* except $CO_2$ concentrations set to 4 times preindustrial | All 1850 except $CO_2$ prescribed at 4 times preindustrial concentration | Same as piControl | Same as piControl | | 1 | 30 | Quantify ERF of $4 \times CO_2$ |



### 3.1.1 Spanning CO2 concentration- and emission-based simulations

Given the increased prominence of science applications for coupled carbon-climate ESMs in climate stabilization and their implications for carbon budgets (Sanderson et al., 2024), the CMIP7 protocol has been re-designed to encourage participation with models driven with both $CO_2$ emissions as well as the more traditional specified $CO_2$ concentrations. The following guidelines seek to maximize comparability between the two sets of simulations:

For models running only with historical $CO_2$ concentrations (i.e. models that run *historical* only):

• run the *historical*, *abrupt-4xCO2*, and *1pctCO2* experiments, branching from year 100 or later of *piControl*.

• the requested length of *piControl* is enough to allow for comparison to all perturbations including future projections and extensions (if applicable) i.e. *piControl* should be as long as the longest perturbation experiment performed.

For models running with BOTH historical $CO_2$ concentrations and emissions (i.e. models that run *historical* and *esm-hist*):

• run the *esm-hist* experiment, branching from year 100 or later of *esm-piControl.*

• the requirements for concentration-driven experiments (*piControl*, *historical*, *abrupt-4xCO2 and 1pctCO2)* as above.

For models running with historical $CO_2$ emissions but NOT planning to run with historical $CO_2$ concentrations (i.e. models that run *esm-hist* only):

• run the *esm-hist* experiment, branching from year 100 or later of *esm-piControl.*

• run the *piControl*, *abrupt-4xCO2* and *1pctCO2* experiments, branching from year 100 (or later, as per modelling

center's preference) of *esm-piControl* with $CO_2$ concentrations as specified in Table 1, but using a pre-industrial value derived from the *esm-piControl* experiment (as discussed in the next paragraph).

Within these general guidelines to accommodate both $CO_2$ emission- and concentration- driven simulations within the same experimental protocol, the CMIP Panel acknowledges that some flexibility in implementation remains necessary. For example, one approach to specifying $CO_2$ concentrations for *piControl*, *abrupt-4xCO2* and *1pctCO2* would be to take the average of the

30 years (i.e. years 70-99) of esm-*piControl*, with *abrupt-4xco2* and *1pctCO2* $CO_2$ concentrations also defined relative to the same level. Another approach could be to preserve model 3-D diurnal to seasonal spatial and temporal variability when forced with $CO_2$ concentrations. Additionally, some centers apply $CO_2$ concentration forcing as a restoring term to the internal atmospheric tracer with a 1/year time scale (Dunne et al., 2020). As background, guidance is that modelling centers should seek to match the observed $CO_2$ concentration in 1850 in their *esm-piControl* and the historical $CO_2$ trend in their *esm-hist*

within ± 5ppm, with larger differences worthy of attention.

### 3.1.2 Historical forcing data sets

Standardized data used to drive simulations has been referred to within CMIP as "forcings" (Durack et al., 2018). This includes specified values of certain variables (e.g. greenhouse gas concentrations) and/or fluxes at domain boundaries (e.g. emissions of carbon dioxide), depending on the experimental protocol. Forcing datasets to be used in the *historical* simulation are



summarized in Table 2. Key changes with respect to CMIP6 include revisions of solar spectral partitioning and geomagnetic referencing (Funke et al., 2024), incorporation of revised satellite and ice core records of historical volcanic activity (Aubry et al., 2021, Chim et al., 2023), comparability of regional emissions of short-lived climate to observations (Hoesly et al., 2023), refined land-use harmonization (Chini et al. 2023). The end of the historical period for CMIP7 is 2022, driven by increased uncertainty in more recent estimates in emission of short-lived climate forcers. These and other forcing improvements will be described in the GMD Special Issue on Forcings as they become available. Models capable of interactive open biomass burning emissions of $CO_2$ are encouraged to run with these emissions interactive rather than prescribed from the available datasets except for $CO_2$ in all concentration-driven runs where $CO_2$ must be explicitly prescribed (*piControl*, *1pctCO2*, *4xabruptCO2*, and *piclim* experiments).

**Table 2: Historical forcings by dataset, provider, short description, temporal range, and documentation. Further details on forcings are provided in papers in a separate collection of GMD/ESSD special issue. Note that modeling centers can chose between CO2 concentrations or emissions from the DECK suite of forcings depending on the simulations. Specification of all the other forcings remains the same between the two types of runs. See https://input4mips-controlled-vocabularies-cvs.readthedocs.io/en/latest/dataset-overviews// as a landing point for modelling teams and https://github.com/PCMDI/input4MIPs_CVs for guidance on current versions of forcings.**

| Forcing dataset | Provider | Short description | Temporal range |
|---|---|---|---|
| **Anthropogenic short-lived climate forcer (SLCF) and CO₂ emissions** | Steven Smith, Rachel Hoesly (PNNL, USA) | Gridded monthly mean historical emission estimates by sector, and fuel for anthropogenic aerosol and precursor compounds, and $CO_2$, $CH_4$ and $N_2O$. | 1750-2022 |
| **Open biomass burning emissions** | Margreet van Marle (Deltares, Netherlands), Guido van der Werf (WUR, Netherlands) | Gridded monthly estimates of open biomass burning emissions (forests, grasslands, agricultural waste burning on fields, peatlands). | 1750-2022 |
| **Land use** | Louise Chini, George Hurtt (University of Maryland, USA) | Gridded annual estimates of the fractional land-use patterns, underlying land-use transitions, and key agricultural management information. | 850-2023 |
| **Greenhouse gas historical concentrations** | Zebedee Nicholls, Malte Meinshausen (University of Melbourne/Climate Resource, Australia) | Consolidated data sets of historical atmospheric (volume) mixing ratios of 43 greenhouse gases and ozone depleting substances. | 1-2022 |



| Stratospheric volcanic SO₂ emissions and aerosol optical properties | Thomas Aubry (University of Exeter, UK), Anja Schmidt (DLR, Germany), Mahesh Kovilakam (NASA, USA) | Timeseries of volcanic aerosol optical properties and volcanic $SO_2$ emissions. | 1750-2023 |
|---|---|---|---|
| **Ozone concentrations** | Michaela Hegglin (Forschungszentrum Jülich, Germany), David Plummer (Environment Canada, Canada) | This is to be determined but the expectation is that it will be - Gridded monthly mean 3-D ozone mixing ratios. | 1850-2022 |
| **Nitrogen deposition** | | This is to be determined but the expectation is that it will be - Gridded monthly mean 2-D nitrogen deposition flux. | 1850-2022 |
| **Solar** | Bernd Funke (IAA, Spain) | Daily and monthly mean reconstructed spectral solar irradiance (SSI) for spectral bins covering the wavelength range 10 – 100,000 nm. | 1850-2023 |
| **AMIP sea surface and sea ice boundary forcing** | Paul Durack (PCMDI/LLNL, USA) | Merged SST and sea ice concentration based on UK MetOffice HadISST and NCEP OI2 | 1870-2022 |
| **Aerosol optical properties/MACv2-SP** | Stephanie Fiedler (GEOMAR, Germany) | Anthropogenic aerosol optical properties for a number of key plumes based on the MACv2-SP parameterization over the 1850-2022 period. | 1850-2022 |


### 3.1.3 Preindustrial control forcing

Forcings for the *piControl* experiment seek to establish a baseline climate against which the forced response can be assessed. The approach in CMIP7 follows CMIP6 although current forcing datasets are to be used. Greenhouse gases, anthropogenic and biomass burning aerosols, and land use forcing use constant 1850 values. Solar forcing uses a fixed mean over two solar

cycles i.e. the average over 1 January 1850 to 28 January 1873 and volcano aerosol forcing for models that prescribe optical



properties use the long-term historical 1850-2022 average values of the historical forcing dataset (Table 2, see also Aubry et al., 2021 and Chim et al., 2023). Averaging is motivated by the observation that multiannual discrepancies in volcanic or solar forcing between *piControl* and *historical* and/or *esm-hist* simulations can lead to drifts (Gregory et al., 2013; Fyfe et al., 2021). Files with the correctly averaged solar and volcanic forcing are provided.

## 3.2 Ocean and Land Spin-up – characterizing model diversity

Prior to starting a control experiment, climate and Earth System models must be integrated to an initial state. This aspect of climate modelling has not traditionally been an issue for weather forecasting because atmospheric dynamics and physics has a relatively short memory of a couple of weeks. Climate, however, has long time scales out to millennia involved in reaching equilibrium in both land (Sentman et al., 2011) and ocean (Irving et al., 2021; Séférian et al., 2016). The CMIP7 protocol described above, as with previous iterations, has no specific requirements for spin-up because the diversity of approaches to developing and spinning up pre-industrial simulations before finalizing the initial conditions for their formal year 1 of the *piControl* mean that it would be difficult at this current moment to specify one amenable to all anticipated participants. Participants are encouraged to provide detailed descriptions of their spin-up methodology and to monitor global energy, water and salinity e.g. via the integrated metrics listed in Appendix 1 and/or save the metrics from the *piControl* data request.

## 3.3 Support for community driven science

CMIP6 supported broad community engagement by soliciting proposals from self-organized MIPs, many of which had long histories themselves. Twenty-two MIPs were eventually endorsed (https://www.wcrp-climate.org/modelling-wgcm-mip-catalogue/cmip6-endorsed-mips-article) and contributed to the CMIP6 request for data. As noted above, this centralized approach required synchronization of the diverse activities represented by the MIPs with the provisioning of forcing and harmonization of the data request on a single timeline set by IPCC AR6.

CMIP7 supports community driven model intercomparisons by providing baseline simulations for comparison, forcing data sets, technical specifications, centralized and distributed infrastructure to access data, and standardized open data access to facilitate model comparison. In CMIP7, however, the CMIP Panel will not endorse entire MIPs but instead draw on the existing experiments designed by community MIPs to assemble compact, targeted collections of simulations to address specific needs. This change is intended to reduce the burden on modelling centers and community MIPs to deliver experimental designs and simulations on IPCC timelines. CMIP7 will thus move to a continuous approach of community MIP contributions supporting novel coupled model intercomparisons. The CMIP Panel, WGCM Infrastructure Panel, infrastructure providers, and IPO will provide support to allow the community MIPs to bring fresh questions, hypotheses, and insight for new experiments, constraints, and applications.

A broad spectrum of modes is available for community MIPs. They may be tightly coupled to CMIP7, for example submitting standardized data to the Earth System Grid Federation; or less tightly constrained by but compatible projects perhaps reusing standards or protocols, or activities which operate completely independently such as nationally and regionally supported





research projects outside the auspices of WCRP. In the absence of centralized endorsement and harmonization of individual MIPs, the CMIP Panel and CMIP IPO play a community service role. This includes encouraging best practices in effective

experimental design and execution through registration and offers guidelines on how best to developing and running MIPs to conform with CMIP Practices in Appendix 2.

**3.4 AR7 Fast Track Experiments**

The AR7 Fast Track is a set of recommended CMIP7 simulations drawn from Community MIPs intended to support both the direct needs of the climate assessment as well as downstream climate services applications from the impacts, mitigation and

adaptation communities, such as the ISIMIP and VIACS initiatives and contribute to the development of high temporal resolution forcing for regionally tailored information through dynamical and statistical downscaling efforts, such as CORDEX. These first focused set of priority recommendations for CMIP7 simulations include: *near-term prediction and long-term projection* experiments that support both the direct needs of the climate assessment as well as downstream use in climate services applications including providing data satisfying the needs of the impacts, mitigation and adaptation communities such

as ISIMIP and VIACS as well as high temporal resolution forcing for regionally tailored information through dynamical and statistical downscaling such as CORDEX. CMIP7 goals also include the more classical aspects of systematic assessment with respect to *characterization* of model diversity, *attribution* of the quantitative role of particular mechanisms in driving the forced response, and *process understanding* as per the four Guiding Research Questions detailed above.

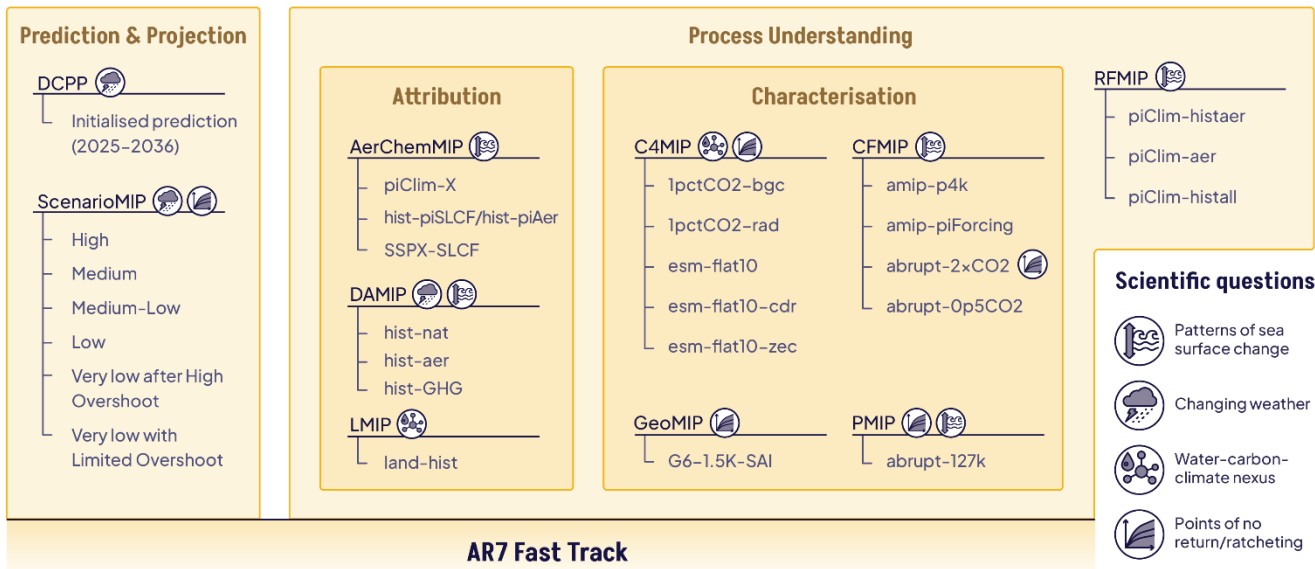

**Figure 3: Schematic mapping the four Guiding Research Questions (Patterns of sea surface warming, Changing weather, water-carbon-climate nexus, and points of no return/ratcheting) and four topical areas (Prediction and Projection, Attribution, Characterization, and Process Understanding) onto AR7 Fast Track experiments.**



### 3.4.1 Harmonization to projections

As in previous phases of CMIP, attention to optimize continuity, or "harmonization" of forcings is necessary across transition
from the end of the historical forcing period heavily constrained by observations (Dec 2021 for CMIP7) into projected future
scenarios from integrated assessment models through ScenarioMIP (van Vuuren et al., in press). The Forcings Task Team's
harmonisation sub-group is working with the ScenarioMIP team on the details of this process, which will be finalised in early
2025. The specification of natural forcings in ScenarioMIP simulations include a projected solar cycle (Funke et al., in
preparation) and a nine-year linear return to the constant background value for volcanoes of [0.013 at 550 nm] as in the
*piControl*.

### 3.4.2 Prediction and projection

Prediction experiments in the Decadal Climate Prediction Project (DCPP) and projections in ScenarioMIP provide important
bounds on a range of possible near-term and future climate outcomes.  While efforts aligned to DCPP exist as an ongoing
effort outside of CMIP as the WMO Global Annual to Decadal Forecast (WMO Global Annual to Decadal Climate Update |
1 | World Meteorological Organization), there is great interest in generating an AR7 "snapshot" of decadal prediction
ensembles that would include a comprehensive suite of model diagnostics consistent with CMIP data standards beyond the
five variables currently made available.

In each previous iteration of CMIP the set of projection experiments included at least one high emissions scenario — initially
viewed as the 1% idealized $CO_2$ increase (IPCC, 1992), then as a "business as usual" (SRES), then as an emissions-intensive
scenario (RCP, SCP), and more recently as a mitigation policy failure scenario (AR6, WGIII Chapter3) — along with a range
of emissions and concentrations scenarios based on moderate to extreme mitigation policy success. Projection scenarios are
being re-envisioned for the AR7 Fast Track by the ScenarioMIP community in close coordination with the CMIP Panel and
WCRP. The focus of this effort is to improve scenario practical viability and comprehensiveness as well as changing the
reference frame from a previous generation emphasis on the null hypothesis of a high emission "business as usual" to the
"current policy" framework developed through the IPCC Working Group III 6th Assessment informed by the Paris Agreement
and ongoing Global Stocktake (https://unfccc.int/topics/global-stocktake). In this reference frame, "current policy" keeps
emissions roughly similar to present-day out to 2100 and provides for a convenient null hypothesis relative to high emissions
"policy failure" versus lower emissions "mitigation policy success" futures (Riahi et al., 2022; Meinhausen et al., 2024).  See
van Vuuren et al. (in press) for a comprehensive discussion of these pathways and their technical implementation into
projections out to 2100 and extensions to 2500.

### 3.4.3 Attribution

One of the key aspects of ongoing CMIP efforts in systematic characterization of model behavior and its relationship to
observations is in attributing the climate response to particular forcing changes, e.g., AerChemMIP and RFMIP for



understanding how individual gases and aerosols affect the energy budget and DAMIP to quantify how different forcings
influence climate. These experiments include a combination of single forcing changes and mechanism withdrawal experiments
that allow for both the quantification of the impact of individual drivers and the combined responses to explore nonlinearity.
From DAMIP, the greenhouse gas only, aerosol only, and natural only experiments are prioritized given their broad use in
prior assessment reports. These will provide the opportunity to examine model response to historical forcings between 2015-
2021 as opposed to the projected forcings used in CMIP6. They will also provide the opportunity to examine the modelled
response to updated forcings prior to 2014, since such differences in forcings can impact on the representation of the historical
climate evolution in individual models (e.g., Fyfe et al. 2021; Holland et al. 2023; Chemke and Coumou, 2024). Comparison
of coupled historical simulations with those in LMIP (and AMIP) allows for attribution of component level biases.  The
increasing use of models with fully interactive carbon cycles also opens the door to attribution of historical changes to
emissions (as opposed to concentrations) and to understand the impact of individual forcings within the context of an
interactive carbon cycle.

### 3.4.4 Characterization

This set of experiments similarly characterizes model ensemble systematic behavior towards understanding why models
produce different outcomes and includes CFMIP for cloud feedbacks, C4MIP to assess carbon cycle-climate feedback strength,
GeoMIP to assess requirements and impacts of purposeful climate modification, and LMIP for the most direct comparison of
land models with observations. As an example of the purpose and interconnectedness of all experiments, an example is
provided for RFMIP that seeks to reduce the large uncertainty in effective radiative forcing due to aerosols in both observations
(Bellouin et al., 2020) and across models (Smith et al. 2020). Experiment *piClim-aer* characterizes the model-specific effective
radiative forcing at present-day (end of *historical*, or 2021 for CMIP7). Understanding of present-day effective-radiative
forcing is augmented by experiments *piClim-histall* and *piClim-histaer*, small ensembles of atmosphere-only simulations with
fixed sea surface temperatures and sea ice concentrations, to characterize the time-varying effective radiative forcing over the
course of the historical period from all natural and anthropogenic forcings and from the temporal evolution of aerosols alone.
Further detail on the motivation for each experiment and context within the MIP from which it is derived is provided in Table
3.

### 3.4.5 Process understanding

The AR7 Fast Track experiments (Table 3) promote the generation of ensembles with complementing available dimensions of
experiment versus structure versus resolution versus ensemble size (Figure 1) to help distinguish the role of different processes
and interactions and local versus remote drivers. Links between the guiding research questions (Section 2) and DECK and
AR7 Fast Track experiments include the following:

- Exploration of the patterns of sea surface warming and changing weather is supported through the updated and
extended AMIP and historical experiments included in the DECK, set of projections and near-term predictions and



associated diagnostics in Decadal Climate Prediction Project (DCPP), Cloud Forcing (CFMIP) and Radiative Forcing (RFMIP) experiments. CMIP and RFMIP also allow exploration of atmospheric feedbacks and identify the role of SSTs in historical evolution and idealized response to forcing. Paleoclimate MIP (PMIP) allow exploration of SST responses to orbital forcing. The single forcing experiments proposed through DAMIP can also help in interpretation

of the role of individual forcings in regional historical trends. The linearity of modelled responses to rising $CO_2$ and feedbacks can also be assessed through comparison of the CFMIP *abrupt-2xCO2* with *abrupt-0p5CO2* experiments. One particularly exciting application of the *esm-flat10-zec* experiment is the ability to conduct long simulations under climate stabilization to develop better understanding of the statistics of climate extremes.

- The *Water-Carbon-Climate Nexus* can be explored through ScenarioMIP projections, Coupled Climate Carbon-Cycle
(C4MIP) and Geoengineering (GeoMIP) experiments. Some of the most societally pressing questions include implications of coupled carbon-climate interactions under a variety of carbon emissions trajectories, particularly under scenarios of climate mitigation (e.g., Carbon Dioxide Removal), interactions of short lived climate forcers under $CH_4$, $H_2$, and greenhouse gas and aerosol emissions trajectories, and advancing process understanding of Earth's radiation budget under purposeful climate modification (e.g., Solar Radiation Management). A series of
idealized diagnostic "flat10" experiments in CMIP7 fast track will be used to derive emissions-driven estimates of Transient Response to Cumulative Emissions (TCRE; *esm-flat10*), Zero Emissions Commitment (ZEC; *esm-flat10-zec*) and climate reversibility under declining to negative emissions (*esm-flat10-cdr*; Sanderson et al. (in review)).

- Tipping *Points of no return/ratcheting* can be explored through both the ScenarioMIP projections (*High*, *Medium*, M*edium-low*, *Low*, *Very Low after High Overshoot*, and *Very Low after High Overshoot*) and extended suite of
idealized response to constant (*esm-flat10*), zero (*esm-flat10-zec*), and declining to negative (*esm-flat10-cdr*) emissions. Another particularly exciting application of the *esm-flat10-zec*experiment is the ability to conduct ensembles of simulations under climate stabilization to develop better understanding of the likelihood of tipping points. The PMIP *abrupt-127k* experiment which allows comparison to model response to last interglacial orbital parameters at which Arctic was free of sea ice and temperatures were close to present-day at preindustrial $CO_2$.


Note that for all AR7 Fast Track experiments that require a historical, present day, or scenarios, the CMIP7 protocol requires slight modification of the original CMIP6 experimental design to be updated to CMIP7 historical (Section ) and ScenerioMIP (van Vuuren et al., in press) forcing.



**Table 3: Overview of the AR7 Fast Track set of experiments with experiment name, experiment primary goal, MIP short name from which it is derived, required model components, brief experiment overview description, primary goal of combined experiments in the MIP from which it is derived, minimum number of years per experiment, and its main purpose. Forcings include Greenhouse Gases (GHG), Short Lived Climate Forcers (SLCFs), Aerosols (AER), and carbon BioGeoChemistry (BGC). Superscripts on the MIP indicate applicability of the experiments to the guiding research questions (Section 2) of 1) Patterns of sea surface warming, 2) Changing extremes, 4) The Water-Carbon-Climate nexus, and 4) Points of no return/ratcheting**

| Experiment short name | Primary Goal of Experiment | MIP short name | Required model components | Experiment overview | Primary Goals of MIP | Years of simulation | Citation for protocol |
|---|---|---|---|---|---|---|---|
| | | | | Prediction and Projection | | | |
| Initialized prediction (2025-2036) | Predicting and understanding forced climate change and internal variability up to 10 years into the future | DCPP[2] | AOGCM | Initialized Prediction: 2025-2036 | Predicting and understanding forced climate change and internal variability up to 10 years into the future through a coordinated set of skillful decadal predictions. | 10 x 10 = 100 coupled | Boer et al., 2016 |
| *High, H-ext* | Climate policy roll-back scenario with low renewable technology development and high emissions and extension to 2300-2500 | ScenarioMIP[2, 4] | AOGCM | Future project simulations out to 2100 and extensions out to 2300-2500 representing mitigation pathways of current policy, | (a) Facilitating integrated research on the impact of plausible future scenarios over physical and human systems, and on mitigation and adaptation options; (b) addressing targeted studies on the effects of particular forcings in collaboration with other MIPs; (c) help quantifying projection uncertainties based on multi-model ensembles and emergent constraints. | 79 + 200+ = 279+ coupled | van Vuuren et al., in press |
| *Medium* | Current policy scenario without further strengthening or roll-back | | | | | 79 coupled | |



| | | | | | | | |
|---|---|---|---|---|---|---|---|
| *Medium-Low, ML-ext* | Modest mitigation policy scenario short of meeting Paris goals and extension to 2300-2500 | | | policy failure, policy success and overshoot. | | 79 + 200+ = 279+ coupled | |
| *Low* | Scenario consistent with staying likely below 2 deg C | | | | | 79 coupled | |
| *Very Low after High Overshoot, VLHO-ext* | Delayed mitigation policy scenario with overshoot but rapidly intensifying CDR to return to 1.5 C and extension to 2300-2500 to return to preindustrial | | | | | 79 + 200+ = 279+ coupled | |
| *Very Low with Limited Overshoot* | Rapid near-term emissions reduction scenario to limit warming to about 1.5 C | | | | | 79 coupled | |
| Attribution | | | | | | | |
| hist-nat | Coupled response to natural solar and volcano forcing | DAMIP [1, 2] | AOGCM | *piControl forcing for all except historically* | (a) Estimating the contribution of external forcings to observed global | 3 x 172 = 516 coupled | Gillett et al., 2016 |



| | | | | *varying solar and volcanoes* | and regional climate changes; (b) observationally-constraining | | |
|---|---|---|---|---|---|---|---|
| hist-aer | Coupled response to anthropogenic aerosol forcing | | | *piControl forcing for all except historically varying aerosols* | future climate change projections by scaling future GHG and other anthropogenic responses using regression coefficients derived for the historical period. (c) Understanding | 3 x 172 = 516 coupled | |
| hist-GHG | Coupled response to anthropogenic GHG forcing | | | *piControl forcing for all except historically varying WMGHGs* | the contribution of individual forcings to inter-model spread over the historical record | 3 x 172 = 516 coupled | |
| land-hist | Evaluate land processes in DECK simulations to identify systematic biases and their dependencies and estimate terrestrial energy/water/carbon variability | LMIP[3] | Land | Update on land forcing modified from ERA5 | Atmospheric reanalysis forced experiment to compare with land satellite and field observations for land model evaluation and benchmarking | 172 land only | Van den Hurk et al., 2016; D. Lawrence, personal communication |
| piClim-aer | Atmospheric response to present-day anthropogenic | RFMIP[1] | AGCM | *preindustrial model SST and SIC and* | (a) Characterizing the global and regional effective radiative forcing for each model for historical and | 30 AMIP | Pincus et al., 2016 |



| | | | | | |
|---|---|---|---|---|---|
| | aerosols to attribute current warming and project committed future warming | | | *forcing for all except 2021 aerosols* | 4xCO2 simulations; (b) assessing the absolute accuracy of clear-sky radiative transfer parameterizations; (c) identifying the robust impacts of aerosol radiative forcing during the historical period. | |
| piClim-histaer | Atmospheric response to historical changes in anthropogenic aerosols to attribute current warming and calibrate emulators | | | *preindustrial forcing for all except historically varying aerosols* | | 30 AMIP |
| piClim-histall | Atmospheric response to historical changes in anthropogenic aerosols and WMGHG to assess why model warming differs from the observed record and estimate model forcing to compare with process models | | | *preindustrial model SST and SIC but otherwise historically varying forcing* | | 30 AMIP |



| piClim-X (where X = Aer, CH₄, NOx, VOC, HC, and N₂O) | Quantifying ERF climate feedbacks for individual SLCFs to assess their contributions to the radiation imbalance. | AerChemMIP[1] | AGCM AER | Single forcing AMIP experiments with model preindustrial SST and SIC | (a) Diagnosing forcings and feedback of tropospheric aerosols, ozone precursors and chemically reactive WMGHGs; (b) documenting and understanding past and future changes in atmospheric chemical composition; (c) estimating their global-to-regional climate response | 30 x 6 =180 AMIP | Collins et al., 2017 |
|---|---|---|---|---|---|---|---|
| *hist-piSLCF (hist-piAer for models without interactive chemistry)* | Diagnosing climate and air quality responses to regionally inhomogeneous evolution of historical SLCF emissions to reduce uncertainty in our understanding of human-influenced climate change. | | AOGCM AER | coupled simulations with historical ly evolving SLCFs | | 172 x 6 =1032 coupled | |
| Process Understanding | | | | | | | |
| *SSPXSST-SLCF (where X = Aer, CH₄, NOx, VOC, HC, and N₂O)* | Quantifying the role of future mitigation actions on SLCFs for climate and air quality responses. | AerChemMIP | AGCM AER | Single forcing experiments in AMIP with model future scenario SST and SIC | As above for AerChemMIP | 30 x 6 = 180 AMIP | Collins et al., 2017 |



| | | | | | | | |
|---|---|---|---|---|---|---|---|
| 1pctCO2-bgc | Idealized biogeochemical response to $CO_2$ concentrations | C4MIP [2, 3, 4] | AOGCM BGC | 1pctCO2 for BGC but pre-industrial $CO_2$ for climate | Understanding and quantifying future century-scale changes in the global carbon cycle and its feedback on the climate system by isolating carbon-concentration and carbon climate elements of the global carbon feedbacks and climate change and enable calibration of coupled carbon-climate emulators. | 150 coupled | Jones et al., 2016; Sanderson et al., 2024; Sanderson et al (in review) |
| 1pctCO2-rad | Idealized radiative response to $CO_2$ concentrations | | | 1pctCO2 for climate but pre-industrial $CO_2$ for BGC | | 150 coupled | |
| esm-flat10 | Idealized coupled response to constant positive $CO_2$ emissions | | | 10 PgC yr-1 $CO_2$ emission | | 100+ coupled | |
| esm-flat10-cdr | Idealized coupled response to reducing positive to negative $CO_2$ emissions after esm-flat10 to diagnose climate response and reversibility after all cumulative anthropogenic emissions are removed | | | $CO_2$ emission declining by 0.2 PgC yr-1 to -10 PgC yr-1 | | 100+ coupled | |
| esm-flat10-zec | Idealized coupled response to zero $CO_2$ emissions after esm-flat10 | | | Zero CO2 emissions | | 100+ coupled | |



| | | | | | | | |
|---|---|---|---|---|---|---|---|
| | to diagnose the Zero Emissions Commitment (ZEC) - the additional warming after the cessation of emissions required to inform remaining carbon budget estimates. | | | | | | |
| amip-p4k | Atmospheric response to idealized ocean warming | CFMIP [1] | AGCM | AMIP SST plus 4K in ice-free regions | Diagnosis of atmospheric response to SST and sea-ice changes for comparison to feedbacks observed and modeled climate sensitivity. | 30 AMIP | Webb et al., 2017 |
| amip-piForcing | Atmospheric response to SST and SIC boundary condition without corresponding forcings | | | AMIP historically varying SST and SIC but preindustrial other forcing | | 30 AMIP | |
| abrupt-2xCO2 | Idealized coupled response to doubled $CO_2$ - similar to 21st century – and in some cases very different from scaled 4x response. | CFMIP [1, 4] | AOGCM | $2xCO_2$ concentration relative to piControl | Improving understanding of circulation, regional-scale precipitation, and non-linear changes. | 300 coupled | Webb et al., 2017 |



| abrupt-0p5CO2 | Idealized coupled response to half $CO_2$ concentration similar to LGM | | | 0.5x$CO_2$ concentration relative to piControl | | 300 coupled | |
| abrupt-0p5CO2 | Idealized coupled response to half $CO_2$ concentration similar to LGM | | | 0.5x$CO_2$ concentration relative to piControl | | 300 coupled | |
| G6-1.5K-SAI | Coupled response to idealized stratospheric aerosol injection to arrest warming to better understand possible consequences of purposeful solar radiation modification | GeoMIP[4] | AOGCM | Stratospheric Sulphur forcing held constant to stabilize climate at 1.5C warming starting from year 2035 of Medium Projection Scenario | Assessing the climate system response (including on extreme events) to proposed radiation modification geoengineering schemes by evaluating their efficacies, benefits, and side effects. | 50 coupled | Visioni et al., 2024 |
| abrupt-127k | Coupled response to orbital changes associated with last interglacial leading to Arctic warming and sea ice loss and translation of high latitude climate forcing to lower latitudes | PMIP[1,4] | AOGCM | Preindustrial forcing except for solar forcing from orbital parameters set for 127K ago | Analyzing the response to forcings and feedback for past climates outside recent variability and assessing the credibility of climate models | 100 coupled | Otto-Bleisner et al., 2017 |



### 3.5 Pre-selection and sub-sampling of models

The number of models contributing results to CMIP has grown substantially over time, such that more than 100 separate models contributed to CMIP6. A dedicated task team (see section 4.1) considered how best to balance insight with a characterization of diversity by considering available approaches to narrow *a-priori* the number of models contributing to each experiment (pre-selection) and the *a-posteriori* narrowing of models having contributed to each experiment (sub-sampling). Many models are closely interrelated, often via shared implementations and more often via shared conceptual bases. Many

sets of simulation results are also similar, though similarity of model structures or genealogies is not an especially good predictor of similarity in results. Similarity between models and between results suggests both that results weighted in some way (e.g. Lee et al., 2021) might be more informative than a raw average, and that having every model contribute to every experiment might be wastefully redundant. The major issues considered and resulting insights drawn from this effort are summarized in Appendix 3.  The end result is that CMIP7 protocols do not include *a-priori* matching of models or

configurations to specific experiments but that CMIP and the Earth System Grid Federation (ESGF) should strive to enhance opportunities for both model and ensemble – based evaluation.  Recommendations for pre-selection and sub-sampling include the following strategies for improving the efficiency of building model diversity in the ensemble:

- Modeling centers consult first the CMIP7 DECK followed by the AR7 Fast Track in their initial prioritization of CMIP7 simulations.

- Researchers interested in assessing the state of the ensemble and Modeling centers wishing to identify gaps in model diversity to further prioritize experiments may consult the proposed Rapid Evaluation Framework (REF; https://wcrp-cmip.org/rapid-evaluation-framework).

- Potential community MIPs looking to build from the CMIP7 DECK and AR7 Fast Track should look at the scope of DECK and AR7 Fast Track experiments supplied on ESGF and analyzed in the REF to identify potential points of

collaboration towards targeted community science goals.

### 4. Evolving CMIP to meet changing needs and opportunities

### 4.1 The CMIP International Project Office

The process leading to this CMIP7 experimental design differs substantially from past iterations of CMIP.  Until CMIP6 the experimental suite was designed almost entirely within the small, researcher-led CMIP Panel relying on volunteer coordination

efforts from individual nations facilitated by WCRP.  While CMIP6 evolved the research scope considerably, CMIP has also been shaped by growing and evolving institutional, national, and international needs for assessment and climate services.  In light of CMIP's widening roles, and in response to the increasing demands of the growing user base, WCRP secured the



establishment of a CMIP International Project Office (CMIP IPO) in 2020 through WMO Resolution 67 (https://www.wcrp-climate.org/images/modelling/WGCM/WGCM23/Presentations/5b_WGCM23-WMO-Res67_CMIP-IPO.pdf). The European

Space Agency successfully bid to host the CMIP IPO, which started operations in March 2022 at their UK site. The provision of full-time staff supports the development and delivery of CMIP consistent with the level of international investment and use. With the IPO in place, the CMIP process is institutionally organized and increasingly consistent with WCRP standards of transparency, inclusiveness, and equity. The IPO also brings the capacity for full documentation of discussions and decisions, coordination of the various panels and task teams (https://wcrp-cmip.org/cmip7-task-teams/) with explicit terms of reference,

and a more open culture allowing many more scientists (including early career researchers) to be engaged.

With staff in place to manage WCRP and stakeholder requests and meeting logistics, the IPO has also enabled more community consultation such as limited term (months to years) task teams formulated to solve particular problems and engage with relevant stakeholder groups. Thus far, seven task teams each involving about a dozen people have contributed to the planning of CMIP7 to date. These include task teams on climate forcings, data access, data citation, data request, model benchmarking, model

documentation, and strategic ensemble design as well as smaller working groups on spin-up, harmonization of historical and projection forcing datasets, thematic diagnostic groups, and sustained mode conceptualization with teams on the CMIP carbon footprint, controlled vocabularies, and quality control/quality assurance being established. The IPO has also facilitated broader community engagement and consultation across the spectrum of time zones with morning, afternoon, and evening virtual information and drop-in sessions plus in person participation in meetings across the globe. While perhaps not as nimble as the

previous small group in-person trusted relationships built over many years, this added layer of organization and associated formality has allowed CMIP to become more transparent, inclusive, and facilitative of robust community engagement.

## 4.2 Maturing infrastructure and support capabilities

The widening use of CMIP data and IPO ability to accumulate, synthesize, and respond to community feedback has underscored several priorities for development of increasing infrastructure and support capabilities in CMIP7. One area of

needed improvement is the uneven availability of model documentation across the ensemble. Downstream users in particular report frustration with descriptions diffused across journal articles, web sites, and technical documents. Modeling centers, on the other hand, are not generally prepared or able to invest substantially in documentation for widespread use. To balance these needs the CMIP7 Model Documentation Task Team has developed a protocol for Essential Model Documentation (EMD): a high-level description of a participating model. It contains information on formulation that can be easily compared

between different models and allows model outputs to be better understood. More detailed model documentation is expected to be available via references given as part of the EMD. Model participation in CMIP7 is contingent on providing this documentation.

The process of collating and reviewing community input into the data request has also been extensively revised. The content of the CMIP7 Data Request starts from a set of Earth System Model Baseline Climate Variables (Juckes et al., 2024) identified

as being of high general utility. This list includes just 132 out 2062 variables in the CMIP6 data request and defines the core



of the CMIP7 Data Request. To enable broader and more transparent access while also scrutinizing and constraining the size and complexity of the request, author teams and scientific steering groups in five thematic areas (atmosphere, ocean & sea-ice, land & land-ice, impacts & adaptation, and Earth system) have been convened with representation from 106 authors from 25 countries). The impacts & adaptation theme has played a critical role in enabling greater engagement with users outside the physical climate science. These teams, working with the CMIP IPO, Data Request Task Team, and WGCM Infrastructure Panel, are consolidating data requirements from MIPs and public consultation into a single comprehensive, or "harmonized" request for the AR7 Fast Track. The CMIP AR7 Fast Track data request will be issued in three major releases, starting with version 1.0 in November 2024 (see https://wcrp-cmip.org/cmip7/cmip7-data-request/).

One of the difficulties in realizing the more routine benchmarking and evaluation of the models as envisioned in Eyring et al. (2016) was the challenges met in automation of diagnostic evaluation – both the difficulty in maintaining and providing to modelling centers the software for doing the analysis and lack of uniformity in metadata and data formats. The Model Benchmarking Task Team has been working to incorporate available open-source evaluation and benchmarking packages into the REF into ESGF to support public analysis and for incorporation into modelling center workflow to internally evaluate developing models in the same comprehensive way before their models are evaluated publicly on these community metrics. Several of such packages widely used in the community for the evaluation and analysis of CMIP6 data have been identified by the task team to provide a "first data check" for newly developed simulations. An overview of the current packages is available on a dedicated subpage of the official CMIP webpage (https://airtable.com/applbQctZtl09L2Ga/shrzOD0Hif0PY6XJI/tbl3p5dTJjQ6xLiWx).

## 5. Summary

CMIP7 continues the pattern of evolution and adaptation to science priorities and community needs building from CMIP6, keeping minimal requirements of DECK and flexibility of infrastructure but switching from officially reviewing and endorsing a broad suite of MIPs in favor of only a targeted set of experiments. CMIP7 science priorities (Section 2) address guiding research questions relating to: 1) Patterns of sea surface warming, 2) Changing extremes, 4) The Water-Carbon-Climate nexus, and 4) Points of no return/ratcheting which are well-aligned with the WCRP's four Science Objectives. The AR7 Fast Track experiments (Section 3.7; Table 3) endorsed in CMIP7 were chosen both to help answer these guiding research questions and address the requirements of prediction and projection (3.7.1), attribution (3.7.2), characterization (3.7.3), and process understanding (3.7.4) towards assessment of state-of-the-art Earth system models.

From consultations with modelling centers and forcings providers, the CMIP Panel anticipates the CMIP7 generation of models and forcings to have improved representation of historical climate changes in addressing CMIP6 deficiencies in the improbably high climate sensitivity (Meehl et al., 2020; Lee et al., 2021) and anomalous cooling in the 1960s (Zhang et al., 2021). The inclusion in HighResMIP2 (Roberts et al., 2024) of models capable of representing tropical cyclones, mesoscale weather systems and eddying ocean interactions brings exciting new potential for characterization of extremes, while the re-



characterization of future pathways into mitigation policy "success" and "failure" relative to "current policy" provides a path for simplifying communication of the Earth system consequences under different policy options. Particularly exciting among

the CMIP7 opportunities is the ability to leverage growing model comprehensiveness and maturity of $CO_2$ emissions-forced ESM's to explore proposed carbon and climate mitigation solutions and the Earth system consequences of stabilization and overshoot and role of changing atmospheric composition and extremes.

CMIP has striven to meet increasing and broadening scientific and service demands, expectations of transparency, diversity, equity, and inclusivity. One dimension of expanded access is Fresh Eyes on CMIP - an early career researcher activity

coordinated through the IPO. To remain sensitive and responsive to the diversity priorities and resource limitations of the modelling canters, CMIP7 provides the revised DECK and AR7 Fast Track recommendations (Section 3) as guidance to modeling centers as they prioritize application of limited computational and human resources. While this initial CMIP7 AR7 Fast Track is aimed at fulfilling the needs of the forthcoming IPCC physical climate and impacts assessment, CMIP also stands ready to consider future targeted sets of experiments developed to suit future needs. This 7th phase of CMIP thus continues at

the heart of internationally coordinated climate and Earth system research within the WCRP and supporting the emerging Climate Service communities.

As the applications of CMIP data continues to widen into new contexts such as machine learning (ML) and new communities including the private sector, the question of assuring "fitness-for-purpose" and the limitations of appropriate use of model contributions grows in importance. There is a growing pressure from stakeholders involved in adaptation and risk mitigation

to provide guidance on appropriate use of individual models and the simulation ensemble as a whole. This is one of the motivations behind CMIP efforts in selection and sub-sampling and deployment of the Rapid Evaluation Framework (REF; Section 3.5; Appendix 3; https://wcrp-cmip.org/rapid-evaluation-framework). Such community pressure will surely grow as emulators based on ML techniques mature and compete with classical physical climate and Earth system models to run large ensembles and downscale information. Emulators may soon enable the construction of more structured ensembles such that a

priori model pre-selection and sub-sampling (Section 3.8) becomes more viable in future phases of CMIP.

CMIP has evolved over its several phases to provide critical services to a broad scientific and stakeholder community through support for protocols including forcing/input data, output conventions, contributions from modelling centers, and mechanisms for data distribution. This chain of end-to-end solutions necessary for coupled model intercomparison is a facility useful for answering a multitude of questions for which CMIP standards, protocols, infrastructure, and experiments provide the basis.

Given this established and ongoing importance of CMIP, it is important to recognize not only the successes but also its ongoing challenges to sustainability. While CMIP has benefited handsomely from the creation of the dedicated IPO, the lack of structural funding for forcings providers, modelling centers, infrastructure providers, and data users forces ad hoc participation based on national funding with diverse priorities. While the effort described above for CMIP in its 7th phase continues as a fundamentally research driven activity, efforts are also underway to build on aspects of CMIP into a more sustained mode.

With the ever-increasing urgency of robust and actionable information for climate change assessment, adaptation and mitigation and predictions on seasonal to decadal timescales, however, the climate community in general (e.g. Schmidt et al.,



2023; Jakob et al., 2023; Stevens, 2024) and CMIP specifically (Hewitt et al., in preparation) have been pursuing ways to support sustained extension of historical forcings, applications of models, and their data provision. CMIP has also identified challenges in the transition of the research mode of funding, human and computational resources, cultures and reward systems
along the path to sustained activity and seeks broad community engagement through WCRP and WMO to continue pressing forward on next generation solutions. These efforts include a recent workshop in October 2024 to explore a "Pathway to regular and sustained delivery of climate forcing datasets" (https://wcrp-cmip.org/event/forcings-workshop/).

In summary, CMIP is evolving to support the ever-increasing diversity of climate and Earth system questions that require a multiverse of models across resolution and comprehensiveness (Figure 1). As this diversity in model structure and applications
expands, CMIP strives to offer a platform that enables intercomparison and hybridization of these approaches to support the international coupled modeling community to understand our present and future climate and their changes and impacts on the Earth system.

**Appendices**

**Appendix 1**

To characterise any model simulation performed before the initial year of piControl (spin-up; Section 3.2), it is recommended that modelling centers save model initial conditions as well as the following integrated annual metrics for provision to the CMIP IPO for curation.

| Metric | Justification |
|---|---|
| TOA radiative imbalance [rsdt, rsut, rlut] | Interpretation of the evolving energy input into the system |
| Global mean SST [tos] | SST stability is essential |
| Ocean heat content – upper and lower if possible [thetaoga, bigthetaoga] | To first order, TOA and ocean heat content change should balance. Upper and lower ocean heat content is preferable – if not total. |
| Total ocean salt content [soga] | Check that the ocean is conserving salt |
| Total ocean mass and volume [masscello, volcello] | |
| Net surface heat flux (into ocean) [hfds, hfcorr] | Check with TOA and heat content (but need to think about ice) |



| | |
|---|---|
| Net surface freshwater flux into ocean and/or global mean precipitation | Check with ocean volume (but need to think about ice) |
| Northern and southern hemisphere sea ice volume/mass min and max [sivoln, sivols] | |
| AMOC [msftyrho, msftyz] | Maximum of MOC in Atlantic |
| Global mean albedo [rsdt, rsut] | |
| Snow cover – total area? [sncls] | |
| CO2mass | Integral of atmospheric CO2 concentration |
| net carbon flux atmosphere-ocean (global integral fgco2) | Understand if any remaining C relocation between the reservoirs is present at the end of spin-up, can be calculated from deltas from total land/ocean/permafrost carbon pools. This can be further detailed.  e.g., Land carbon can be distinct between soil/vegetation/permafrost, ocean carbon can be distinct between DIC/DOC/POC/surface ocean/deep ocean, ... |
| net carbon flux atmosphere-land (nbp) | This may need to be derived if terms like fire and land use are treated separately |
| Net permafrost carbon flux | |
| Sediment weathering flux / riverine C flux (icriver, ocriver, fric, froc) | Necessary for mass balance within the ocean. There are separate terms for inorganic and organic carbon |





| Diagnosed $CO_2$ emissions | In case of $CO_2$ concentration or emissions driven spin-up, respectively, to assess the total C balance of the model. |
|---|---|
| intCVeg | Integral of Carbon in Vegetation (Three of these four land carbon metrics would be useful to track drift in stocks) |
| intCsoil * | Integral of Carbon in soil |
| intCLitter | Integral of Carbon in litter |
| intCLand | Integral of Carbon on Land |
| intdic | Integral dissolved inorganic carbon concentration |
| intAlk | Integral dissolved alkalinity concentration |
| $intO_2$ | Integral dissolved oxygen concentration |
| $intNO_3$ | Integral dissolved nitrate concentration |
| Total water storage | sum of snow water equivalent and soil moisture in all layers, useful to track drift in water budget |

**Appendix 2**

**General guidance on setting up a MIP**

CMIP's long experience in coordinating model intercomparisons has helped identify a set of practices that allow broad participation and efficient use of resources, which are summarized here.

1. Articulate the hypothesis: Clearly define what new knowledge will be gained by the experiments. MIPs that define key metrics that can be calculated and compared with observed quantities are particularly useful in this regard.

2. Clarify the experimental design and data requirements: Experimental designs are most effective when they elucidate areas of robust model agreement and inform on areas of inter-model differences. Clear design and description of a MIP and its individual experiments, articulation of data requirements, and resource planning is essential to ensure



uniform conformance to protocols by contributors and the production of comparable results that meet the design goals. Targeted sizing of the experimental design (in terms of both runs and data requirements) helps limit the environmental footprint of performing the MIP simulations.

3. Leverage past experience: An awareness of previous model experiments and care in avoiding unnecessary duplication frees resources and focuses effort on novel questions. Designs explicitly taking into account the extent to which modestly different forcings, experiments, or model versions can provide compelling motivation for new experiments.

4. Develop prototype experiments: Performing prototype experiments with at least one model prior to proposing MIP experiments provides critical justification of why initial results are insufficient and need to be augmented with results from a multi-model ensemble. Identification of dependencies or links to existing (or proposed) experiments and associated available simulations provides a comprehensive perspective on the full requirements for participation.

5. Foster transparent and inclusive collaboration: MIPs co-designed by a wide range of individuals, communities, and institutions contributing ideas, simulations, results, or analysis help move the field forward. Reaching out early to modelling centers and/or other participants can help secure sufficient commitments to assure the experimental goals can be met. MIPs are encouraged to consider all aspects of diversity (e.g., geographical, gender, career stage) when building their leadership team in line with WCRP goals (see Section 6 WCRP Guidelines on Membership and Responsibilities)

6. Coordinate with other MIPs: Consider registering the MIP. This includes a brief description of initial plans and is meant to identify potential duplications and foster opportunities to coordinate across MIP activities. Such coordination is particularly helpful for avoiding naming clashes, which can create confusion for modelling teams and downstream data users alike.

7. Document the approach comprehensively: Description papers subject the MIP design to a process of peer review. Such papers provide the goals of the MIP and the rationale for each of the planned experiments. Defining the experiment protocols as clearly as possible helps avoid confusion and highlight possible areas of departure between modeling center implementation. "Living" experiment documentation on a website or other easily accessible platform can ensure that up-to-date information is readily available for those seeking to conduct the experiments.

8. Prioritize anticipated experiments: Explicit prioritization ("tiers") of experiments allows contributors to usefully participate at whatever level of effort best suits them for a spectrum of levels of engagement.

9. Support contributors and users: Anticipate how the data will be prepared and distributed so that the scientific findings can be published including testing diagnostics across models to assure data comparability.

10. Acknowledge contributions: Where MIP analysts are distinct from the groups contributing results encourage inclusion of data providers as co-authors (especially in early publications). Data citation is a further mechanism of acknowledgement.



**Conforming with CMIP Practices**


In addition to following the above "best practices", a MIP may want to take advantage of the data standards and infrastructure that support the most recent phase of CMIP. In some cases, the CMIP panel and IPO may be able to provide additional input and services that may increase the potential scientific impact of a MIP. Insistence on the latest standards and adoption of the same controlled vocabularies used in previous CMIP phases can reduce the overhead on modeling group participation and

facilitate community analysis of MIP results. While the CMIP7 technical specifications are still under development, they will rely heavily on the CMIP6 requirements which were discussed generally in Balaji et al. (2018) and were fully detailed on the CMIP6 website in the Guide to CMIP6 Participation.

**Appendix 3: Model sub-selection**

Noting that the number of models contributed to CMIP has grown substantially from CMIP3 to officially over 100 models in CMIP6 and that the computational, energy, and human resources available for CMIP-related activities is limited, the design phase for CMIP7 explored options for sub-sampling the ensemble by pre-selecting models for individual experiments with an eye towards optimizing computational efficiency. The final design, however, does not include a pre-selection of models. The reasons for this decision are laid out in this appendix.

Support for pre-selection of models comes from several bases, including the recent weighting of CMIP6 model output conducted in multiple studies and applications. One of the important departures of the IPCC 6th Assessment from those previous was a shift towards a synthesis of multiple lines of evidence to inform future climate uncertainty ranges (using a combination of ESM ensembles, observations and emulators). This, in part was due to a subset of models which were deemed to exhibit historical warming which was inconsistent with observations (Hausfather et al 2023). Potential mechanisms for

direct model weighting on global warming response have been proposed by some authors (Massoud et al 2023), while others propose multivariate weighting of models based on aggregate skill and independence (Sanderson, et al 2017, Brunner et al 2020 ). It is also recognized in extensive literature (Knutti et al., 2013) that the diversity of current models arises from a smaller number of lineages which maintains dependency between them in the algorithmic structure and behavior (e.g., CESM to NorESM, E3SM, CCMC, BCC-CSM), which some studies have recommended as a strategy for weighting (Kuma et al. 2023).

There are also several strong arguments against pre-selection of models. In many cases, despite their common ancestry, seemingly similar models can behave very differently. For example, in CMIP6, the atmospheric component of NorESM2 is very close to that of CESM2, yet CESM2 had one of the highest equilibrium climate sensitivities at 5.2K and NorESM2-LM had one of the lowest at 2.5K (Meehl et al. 2020, Table 2). Results from Perturbed Parameter Ensembles also demonstrate that small changes in parameter tuning can yield strongly differing results from the same model (Yamazaki et al. 2021), which

makes it challenging to determine how to balance ensuring independence with spanning as broad a range of uncertainty space as possible. While many models participating in CMIP include different configurations of the same trunk model (ESM, high



resolution, alternative physics), this potential source of duplicity often provides valuable dimensions of diversity include not only the most comprehensive and high-resolution models but also more computationally efficient models which generally participate in targeted community science activities within CMIP. Further, even if it is feasible to choose the "best" models for a particular task, there are several benefits to a diverse ensemble which spans a wide range of plausible behavior. Insights into mechanisms and constraints on future projections such as "emergent constraints" benefit from the full range of responses that can allow linkages between aspects of the model representation and forced response to be identified. Model spread in future climate response cannot be not known in advance, and only in ensemble post-processing is it evident how process and technical improvements translate into ensemble performance and projection spread. While immensely valuable in combining multiple lines of evidence to constrain the global temperature response once the ensemble is mature, these approaches cannot be used a priori to select models to participate in CMIP experiments because model simulations are not yet available, making objective pre-selection of CMIP7 model variants effectively impossible. Further, such techniques are highly dependent on the metric chosen - two models may exhibit highly similar warming patterns, but different precipitation or carbon cycle responses - for example. Any attempt to pre-select independent models would require a highly multivariate approach. Studies such as Peatier (2023) and Sanderson (2017) also suggest that as the number of metrics included in an assessment increase, the ability to distinguish skill and similarity in that space weakens (even post-hoc) such that the more metrics are considered, the less significant the differences between models becomes in terms of overall performance and the more arbitrary the weighting. As such, it is not desirable to filter potentially useful and unique models until their historical performance and basic metrics of future climate response are known.

In contrast, post-selection and model weighting strategies have proven immensely useful for downstream and targeted community science activities which are able to select models based on simulations in the CMIP7 DECK and AR7 Fast Track in cases when desired diagnostic behavior is well defined. There are several examples of frameworks developed through CORDEX for sampling based on metrics for different regions (e.g., Grose et al., 2023, Nguyen et al., 2024). In many cases, however, these configuration-specific model variants are already effectively designed for specific parts of CMIP (e.g., high resolution for HighResMIP, interactive chemistry for AerChemMIP, interactive carbon cycle for C4MIP).

In the absence of pre-selection modeling centers might help fill uncertainty space by consulting results from the Rapid Evaluation Framework (REF), identifying gaps in model diversity across dimensions such as CO2 and aerosol sensitivity, temperature and precipitation bias patterns, carbon response patterns, etc., and contributing simulations to fill uncertainty space towards yielding new information to robustly fill out the ensemble.

**Code availability**

While no code was used in the present study, CMIP best practices are for modeling centers to make the code for models used for the DECK and AR7 Fast Track simulations described here be made available as part of the documentation of their models.



**Data availability**

While no data was used in the present study, the model output from the DECK and AR7 Fast Track simulations described in
here will be distributed through the Earth System Grid Federation (ESGF). As in CMIP6, the model output with associated
metadata and documentation will be freely accessible through data portals.

**Author contribution**

JD prepared the manuscript with contributions from all the co-authors.

**Competing interests**

The authors declare that they have no conflict of interest.

**Acknowledgements**

The CMIP IPO is hosted by the European Space Agency, with staff provided on contract by HE Space Operations Ltd. Many
individuals contributed substantively to ongoing discussions as part of the various CMIP7 task teams including Sasha Ames,
Thomas Aubry, Ben Booth, Laurent Bopp, Dong-Hyun Cha, Louise Chini, Elizabeth Dingley, Daniel Ellis, John Fasullo,
Stephanie Fiedler, Bernd Funke, Matthew Gidden, Heather Graven, Michael Grose, Tomohiro Hajima, David Hassell,
Michaela Hegglin, Rachel Hoesly, Forrest Hoffman, Jarmo Kikstra, Andrew King, Jean-François Lamarque, Guillaume
Levavasseur, Mahesh Kovilakam, Thibaut Lurton, Chloe Mackallah, Claire Macintosh, Ken Mankoff, Margreet van Marle,
Malte Meinshausen, Nadine Mengis, Atef Ben Nassar, Swapna Panickal, David Plummer, Keywan Riahi, Bjørn Samset,
Roland Séférian, Anja Schmidt, Chris Smith, Doug Smith, Steven Smith, Abigail Snyder, Christian Steger, Tim Stockdale,
Martina Stockhause, Abigail Swan, Briony Turner, Detlef van Vuuren, Guido van der Werf, and Tilo Ziehn and the many
individuals across the CMIP7 task teams whose details can be found at https://wcrp-cmip.org/cmip7-task-teams/.

**Financial support**

OB was supported from the CLIMERI research infrastructure and the Agence Nationale de la Recherche - France 2030 as part
of the PEPR TRACCS programme under grant number ANR-22-EXTR-0001. HH was supported by the Met Office Hadley
Centre Climate Programme funded by DSIT. IRS was supported by the NSF National Center for Atmospheric Research, which
is a major facility sponsored by the NSF under Cooperative Agreement No. 1852977. ZN acknowledges funding from the
CMIP IPO, hosted by the European Space Agency, and the European Union's Horizon 2020 research and innovation
programmes (grant agreement no. 101003536) (ESM2025). BH acknowledges funding from the European Union's Horizon



2020 research and innovation programmes (grant agreement no. 101003536) (ESM2025). JA acknowledges support from the
ARC Centre of Excellence for the Weather of the 21st Century (CE230100012). The work of PJD and KET from Lawrence
Livermore National Laboratory (LLNL) is supported by the Regional and Global Model Analysis (RGMA) program area under
the Earth and Environmental System Modeling (EESM) program within the Earth and Environmental Systems Sciences
Division (EESSD) of the United States Department of Energy's (DoE) Office of Science (OSTI). This work was performed
under the auspices of the US DoE by LLNL under contract DE-1175 AC52-07NA27344. LLNL IM Release: LLNL-JRNL-
835  1109530

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
