# Peer review of "An evolving Coupled Model Intercomparison Project phase 7 (CMIP7) and Fast Track in support of future climate assessment"

_EGUsphere, 2024_

## Referee Comment (RC2)

Firstly to say that the CMIP panel and authors here are to be congratulated on the way they have approached the task of developing CMIP7 plans in a complex landscape of requirements. CMIP has had a lot of success historically but requirements have grown and that growth is not sustainable so the new approach to consult with both users and providers and hence prioritise a more manageable, but still vital, set of simulations has been extremely welcome. The outreach, consultation and dissemination of information has been excellent throughout and this paper contributes to that process.

CMIP is a huge undertaking and changes the deployment of resource (both personal and computing/technology) in many, many modelling and research centres around the world. Careful design of what is requested and why is essential.

I perform this review mainly in the context that the main aspects of CMIP7 and the Fasttrack, are already determined and too late to make substantial changes. Therefore, I focus on the presentation and explanation aspects with a few suggestions of things which could still be tweaked or clarified.

**My major comment** is to ask for more details on where the "Guiding Research Questions" came from? Are these the result of a consultation on the priority climate science questions? They resemble, but are not the same as, past WCRP grand challenges (e.g. on extremes or carbon cycle). The way the paper is presented implies you started with these as a guiding set of questions and designed CMIP7 to answer them. But in practice that wasn't how I recall it happening – so have these questions been retro-fitted to the experiments? E.g. line 132 says that CMIP7 design came from consultation and surveys – this is certainly true of the experiments – but did this consultation also take place for the science questions?

When I look over the CMIP7 web page there are lots of details and further links to the experiments, the task teams, the data request, the REF etc. Your figure 3 is replicated on the website, which mentions the science questions linked to each FT experiment - but I cannot see the questions described or explained anywhere. It feels like these questions have been added after the experiment design. If these really are "guiding questions" that have guided, and are intended to keep guiding, CMIP I think they and their derivation need more prominence.

It is not clear, for example, why you identify SST patterns over, say, cloud feedbacks, as a key driver of system sensitivity? Also, when you discuss a "carbon-water nexus" – is this just a catch-all for things not included in the other questions? The paragraphs of description of this question (sec 2.3) don't appear to cover interactions between carbon and water cycles as implied by the "nexus" tag.

So overall it would be good to articulate maybe how these priorities were arrived at. I am not querying the importance of these questions – they are clearly crucial. But other aspects (for example on aerosol forcing and cloud processes) could also be seen as equally important, and CMIP7 will address many more than just these. Maybe it is better to present the experiments first and then give some example high priority questions as examples of things which CMIP7 may help address – but it feels to be overselling the tag of "guiding questions" to imply that these came first and led to the CMIP7 design.

**Other suggestions I think are important:**

**Model/simulation quality**

i.      Lines 374-375 – it feels reasonable to suggest a degree of stability of a control run: +-5ppm is probably OK – but better as a rate than an absolute – is this +-5ppm *per century* for example? In CMIP6 C4MIP requested drifts of less than 10 PgC per century in the main pools. But it would be consistent to also request stability criteria for other metrics – e.g. global T must drift by no more than +- XX degrees, or AMOC within XX Sv. It would be good to treat all major climate components similarly.

ii.     More importantly – I think it is unwise, however, to suggest arbitrary quality criteria for historical runs. Many ESMs may not hit the historical CO2 within 5ppm. See e.g. Hajima et al ([https://egusphere.copernicus.org/preprints/2024/egusphere-2024-188/](https://egusphere.copernicus.org/preprints/2024/egusphere-2024-188/)) for thorough evaluation of CMIP6 models in this respect. What happens if a model does not hit you 5ppm bounds – is it excluded from analysis? Again – as above, will you also specify acceptance criteria on other measures? – e.g. goodness of fit of the historical temperature record? This would be a big change for CMIP – to specify acceptance criteria – I think it needs much more consultation before you introduce this.

**Ensembles** – do you have any recommendations around generation of ensembles (from each model)? I realise you don't want to rule out models by requiring large ensembles, but some experiments may benefit more than others from ensembles. Line 510 says that the FT "promotes the generation of ensembles" – but it is not clear how? FT does not appear to mention ensembles at all – but it could be a good opportunity to do so.

It might be useful to provide guidance on this without mandating. Likewise you could guide on choice of initial conditions (e.g. branch points best taken >XX years apart from the control run).

As an example, quantifying TCRE from flat10 is a relatively large signal-to-noise activity. Ensembles may add little value to this. But quantifying ZEC from the flat10-zec simulation is a very small signal-to-noise and ensembles of this run could be really useful. See e.g. Borowiak et al ([https://agupubs.onlinelibrary.wiley.com/doi/full/10.1029/2024GL108654](https://agupubs.onlinelibrary.wiley.com/doi/full/10.1029/2024GL108654)) which shows that ZEC derived from CMIP6 ZECMIP are subject to a level of uncertainty which CMIP6 did not consider due to lack of ensembles.

**Spin-up.** I'm not sure I understand the request to submit numerical results from the spin-up of the models. What is the goal of this – how will they be used? "for curation" sounds like an odd phrase – why do these need curating? And what does "curation" involve – is this the same as archiving on a public database like ESGF?

**Model selection**. I think you are very wise not to do any prior screening or selection of models. The "hot models" paper you cite in Appendix 3 by Hausfather et al is rather simplistic to provide a table of "Y" and "N" on model screening based on sensitivity. A more nuanced analysis by Swaminathan et

al (https://agupubs.onlinelibrary.wiley.com/doi/full/10.1029/2024EF004901) shows clearly that many metrics of crucial interest are not related to ECS. Many high sensitivity models have very good evaluation scores on many metrics and vice versa – having a lower ECS is certainly not a measure of quality. Any screening or selection needs to be much better understood and carried out case-by-case for the application in question. It cannot (yet) be done at the scale of CMIP which has so many down-stream uses of the outputs.

**Minor comments**

- Lines 102-107. This is a nice description of how CMIP has expanded and refined focus as both the expertise and need evolves. It feels that more knowledge of reversibility and symmetry is a big gap in our understanding of the climate system, and here could be a good place to articulate the need for more process exploration of how the system behaves under reversing of forcing.
- Line 216 says that CMIP7 focus on emissions-driven runs allows for more exploration of extremes under stabilisation – can you explain how so?
- Sec 2.4 on points of no return – is there a reason not to call this either "tipping points" or "irreversibility" which have become much more common phrases for these topics. Wood et al (2023 - https://agupubs.onlinelibrary.wiley.com/doi/full/10.1029/2022EF003369) is a good reference here for the framing of high impact/low likelihood outcomes and the need for research spanning different dimensions of this topic.
- Line 297 onwards – describing the CMIP7 DECK intent. It is worth being explicit here that the goal is only to characterise the response to _increasing_ forcing. It was a deliberate decision not to add a DECK experiment to characterise the system response to reducing forcing. (This remains a gap in CMIP7 – noting that flat10-cdr can only be performed by ESMs)
- Table 1 is important. A couple of notes/suggestions
    - For esm-piControl the forcing is described as "emissions" - I wonder if this should be better described as "interactive CO2" or "simulated CO2" because of course there are no emissions. So even though we informally describe this as "emissions mode" it risks implying that there are some emissions being applied. Or at least specify that CO2 emissions are zero.
    - Typo – looks like the 1% and historical lines have transposed the solar/volcanic forcing entries
- Line 355. Can you clarify the need for 100 years of control run before any experiments are branched off? I don't recall this being requested in CMIP6
- Line 364 – can you explain why conc-driven control run is required if the esm-control is stable? That seems redundant
- Table 2 is useful – but it feels odd to name individuals. What happens as/when a person moves job etc? maybe a named group in an organisation is more useful.
- Table 2, N deposition. Will this be speciated into dry/wet and oxidised/reduced reactive nitrogen?
- Line 405. The section on spin-up – it is not clear how the strap line "characterising model diversity" is relevant to this sub-section. Maybe just call the section "ocean and land spin-up" (where land here includes land ice/cryosphere?)
- Line 470 – is "SCP" a typo? "SSP"?

- Table 3 is super useful and important – it will be a very good easy-look-up of the whole set of FT simulations. But it is really big! It is important that it is produced and typeset to be easily readable given how big it is. I feel this comment may be more for the journal/typesetters than the authors – I hope you can find a way to make it well readable.
- Table 3 – scenario time period. You quote that scenarios run to 2100 – is this decided? I thought it would be 2125, or at least this was still being discussed. (personal opinion – it drives me mad that IPCC figures and values can only ever quote a climate – i.e. 20-year average – for 2090. So an extension to a minimum of 2110 seems vital so that we can actually quote a 2100 value for projected results!)
- Appendix 1 – requested spin-up metrics. As per my comment above I'm not yet convinced why you need to request these. But if you do, then to close the land carbon cycle you should also requested cProduct. Even if the control run has no land-use _change_ it will still have land use, and the product pools may well be non-zero. cLand is then the sum of cVeg+cLitter+cSoil+cProduct

---

## Community Comment (CC1)

**Review of "An evolving Coupled Model Intercomparison Project phase 7 (CMIP7) and Fast Track in support of future climate assessment" by Dunne et al [egusphere-2024-3874]**

**Summary**

The authors motivate and describe the seventh iteration of CMIP, including the new Fast Track set of experiments which serves the IPCC. The paper is mostly effective in achieving these goals, but there are a few areas needing improvement. This review largely deals with issues relevant to the Cloud Feedback Model Intercomparison Project (CFMIP).

Mark Zelinka
Maria Rugenstein
Alejandro Bodas-Salcedo
Jennifer Kay
Paulo Ceppi
Mark Webb
on behalf of the CFMIP Scientific Steering Committee

**Major Comments**

- Section 2.1 describes the first of four guiding questions in CMIP7, dealing with pattern effects. A large part of the reason the scientific community is interested in pattern effects is because of the science conducted by members of the CFMIP community (Andrews et al. 2015; Zhou et al. 2016; Andrews and Webb 2017; Ceppi and Gregory 2017; Andrews et al. 2018, 2022), facilitated by CFMIP experiments like amip-piForcing (Andrews 2014; Webb et al. 2017), and illuminated by CFMIP diagnostics (including satellite cloud simulator diagnostics that reveal the diverse cloud responses to warming patterns). The "Why expect progress now?" section completely excludes a role for CFMIP while instead mentioning the roles that can be played by DAMIP and AerChemMIP. The focus here seems to be more on what causes warming patterns (a worthy goal), but the understanding of the climate response (including but not limited to clouds) to diverse warming patterns is essential to this problem and should not be neglected. Moreover, the surface temperature response pattern is likely to be at least partly affected by how clouds and their radiative effects feed back on warming patterns (Myers et al. 2017; Erfani and Burls 2019; Rugenstein et al. 2023; Espinosa and Zelinka 2024; Breul et al. 2025) and are involved in teleconnections that propagate surface temperature anomalies from high to low latitudes (Kang et al. 2023; Hsiao et al. 2022). We suggest better acknowledging CFMIP contributions to the current understanding of the pattern effect and explicitly calling out the role that CFMIP can play in making progress. We also note

that the first sentence of this paragraph is rather hard to parse and is formulated rather weakly ("xyz may all help" – it remains unclear with what and how).

- CFMIP requests that the abrupt CO2 experiments (4x, 2x, and 0.5x) be run out to a minimum of 300 years, and we strongly encourage modeling groups to run beyond that (which could be noted at L331). Note that CFMIP requested this minimum duration as part of the FastTrack consultation process, which was then adapted into the request for the abrupt CO2 experiments. (See the abrupt-4xCO2 request: [https://airtable.com/embed/appVPW6XAZfbOZjYM/shrqq9I4NJThwOT9W/tblkc1lkKEtiYKcho/viw9PLlrOnfUMcvHw/recl01t59HM8jz8ax](https://airtable.com/embed/appVPW6XAZfbOZjYM/shrqq9I4NJThwOT9W/tblkc1lkKEtiYKcho/viw9PLlrOnfUMcvHw/recl01t59HM8jz8ax).) Table 1 currently lists the abrupt-4xCO2 run as extending for "150+ (300)", though it is not clear what this nomenclature means exactly.  We request that "150+" be replaced with "**300+**" to make it clear that 300 years is the desired minimum, and "(300)" be replaced with "**(1000)**". The reasons for requesting that the abrupt CO2 runs be integrated for a minimum of 300 years with strong encouragement to extend beyond that are manifold:
    - **Better ECS quantification:** Rugenstein and Armour (2021) quantified with 10 equilibrated CMIP5 and CMIP6 models that 400 years are necessary to estimate the true equilibrium climate sensitivity within 5% error. The model spread in equilibration is large and CMIP6/7 models probably need longer to equilibrate due to the "hot model problem" (Hausfather et al. 2022), which partly consists of temperature- and time-dependent feedbacks. Kay et al (2024) estimated an equilibrium timescale of 200+ years for 2xCO2 and 500+ years for 0.5xCO2, noting important implications for paleo cold climate constraints (e.g., LGM) that can only be understood if the simulations are long enough.
    - **Understanding centennial coupled behavior:** Simulations of at least 300 years are necessary for estimating the pattern effect, ocean heat uptake and convection (Gjermundsen et al. 2021), AMOC recovery (Bonan et al. 2022), and Equatorial Pacific response timescales (Heede et al. 2020).
    - **Understanding and quantifying feedback temperature dependence:** This is not well understood, could lead to tipping points and is, after the pattern effect and cloud feedbacks, the biggest unknown in estimating ECS, understanding hot models, and high-risk futures (Bloch-Johnson et al. 2021). It is very hard to quantify because it is obscured by the pattern effect, but is aided by longer simulations.
    - **Practical considerations:** Running existing simulations for longer is typically easier than running new simulations. Thus, if computing time is available at modeling centers, it is strongly encouraged that pre-industrial control and abrupt CO2 runs be extended as long as possible. Anecdotally, many of the model centers contributing to LongRunMIP (Rugenstein et al. 2019) had independently run their simulations for longer than 150 years and had the data sitting around, suggesting that in many cases such long simulations are already being performed or are trivial to extend. Currently, ~52 groups are using the LongRunMIP simulations for studies on internal variability, global warming levels, feedback quantification, paleo climate, oceanography, and training for data-driven machine learning approaches.

**Minor Comments**

- L34: Should it be "...include **experiments to diagnose** historical…"?
- Introduction section: This section may be too long. The main audience of this paper is the science community that want to understand the rationale and details of the experimental design, not the history of CMIP iterations.
- L90: should be Zelinka et al 2020
- L125-127: Suggest being more specific and use "modeling community", rather than "research community" as a whole. The research community benefits as a whole, but it doesn't share the burden.
- L130" "... the present experimental design includes some components …" This point is hard to parse. The entire paragraph reads well though, but the role DECK plays in climate services might need more highlighting. The remainder of the paper is phrased mostly in terms of science questions and the role climate service plays in there remains somewhat unclear.
- L140: Would it be worth listing a few big questions which were answered mainly or only through past CMIP cycles?
- L265-266: something wrong with the phrasing here
- Table 1: It's unclear why the request is for a small ensemble for historical and a large ensemble for amip
- Section 3.1.2: It would be helpful to see a plot of how the new forcing datasets differ from those used in CMIP6 during the 1850-2014 period.
- L310/Fig.2: This schematic might benefit from a vertical time axis. The current version leaves a lot of room for interpretation. What are the small orange arrows? What is the connection between DECK and AR7 Fast Track?
- L355: "year 100 or later of piControl" – is the rationale for this given anywhere in the manuscript?
- L383: The historical and AMIP simulations end in 2021 according to Table 1.
- L498: CFMIP deals with cloud **and non-cloud** feedbacks (all radiative feedbacks)
- L501: Figure 3 excludes RFMIP from the "Characterization" box, yet it is highlighted in this Characterization section, which is confusing.
- L510-511: Very hard to parse this statement
- L516: "Forcing" should be "Feedback"
- L517: I believe you mean "CFMIP" rather than (or in addition to) "CMIP" here
- L541: Missing section number
- Table 3, amip-p4K: missing word here? "feedbacks observed"
- Table 3, amip-p4K: the number of years should be 44 (1979 - 2022)
- Table 3, amip-piForcing: the number of years should be 153 (1870 - 2022)
- L638: 4 should be 3
- Appendix 1 table: Suggest specifying **top of atmosphere** albedo when referencing rsdt and rsut
- L712-713: Might be some missing words here

**References**

Andrews, T., 2014: Using an AGCM to Diagnose Historical Effective Radiative Forcing and Mechanisms of Recent Decadal Climate Change. https://doi.org/10.1175/JCLI-D-13-00336.1.

——, and M. J. Webb, 2017: The Dependence of Global Cloud and Lapse Rate Feedbacks on the Spatial Structure of Tropical Pacific Warming. *J. Clim.*, **31**, 641–654, https://doi.org/10.1175/JCLI-D-17-0087.1.

——, J. M. Gregory, and M. J. Webb, 2015: The Dependence of Radiative Forcing and Feedback on Evolving Patterns of Surface Temperature Change in Climate Models. *J. Clim.*, **28**, 1630–1648, https://doi.org/10.1175/jcli-d-14-00545.1.

——, and Coauthors, 2018: Accounting for Changing Temperature Patterns Increases Historical Estimates of Climate Sensitivity. *Geophys. Res. Lett.*, **0**, https://doi.org/10.1029/2018GL078887.

——, and Coauthors, 2022: On the Effect of Historical SST Patterns on Radiative Feedback. *J. Geophys. Res. Atmospheres*, **127**, e2022JD036675, https://doi.org/10.1029/2022JD036675.

Bloch-Johnson, J., M. Rugenstein, M. B. Stolpe, T. Rohrschneider, Y. Zheng, and J. M. Gregory, 2021: Climate Sensitivity Increases Under Higher CO2 Levels Due to Feedback Temperature Dependence. *Geophys. Res. Lett.*, **48**, e2020GL089074, https://doi.org/10.1029/2020GL089074.

Bonan, D. B., A. F. Thompson, E. R. Newsom, S. Sun, and M. Rugenstein, 2022: Transient and Equilibrium Responses of the Atlantic Overturning Circulation to Warming in Coupled Climate Models: The Role of Temperature and Salinity. https://doi.org/10.1175/JCLI-D-21-0912.1.

Breul, P., P. Ceppi, and P. Nowack, 2025: The importance of stratocumulus clouds for projected warming patterns and circulation changes. *EGUsphere*, 1–22, https://doi.org/10.5194/egusphere-2025-221.

Ceppi, P., and J. M. Gregory, 2017: Relationship of tropospheric stability to climate sensitivity and Earth's observed radiation budget. *Proc. Natl. Acad. Sci. U. S. A.*, **114**, 13126–13131, https://doi.org/10.1073/pnas.1714308114.

Erfani, E., and N. J. Burls, 2019: The Strength of Low-Cloud Feedbacks and Tropical Climate: A CESM Sensitivity Study. *J. Clim.*, **32**, 2497–2516, https://doi.org/10.1175/jcli-d-18-0551.1.

Espinosa, Z. I., and M. D. Zelinka, 2024: The Shortwave Cloud-SST Feedback Amplifies Multi-Decadal Pacific Sea Surface Temperature Trends: Implications for Observed Cooling. *Geophys. Res. Lett.*, **51**, e2024GL111039, https://doi.org/10.1029/2024GL111039.

Gjermundsen, A., A. Nummelin, D. Olivié, M. Bentsen, Ø. Seland, and M. Schulz, 2021: Shutdown of Southern Ocean convection controls long-term greenhouse gas-induced

warming. *Nat. Geosci.*, **14**, 724–731, https://doi.org/10.1038/s41561-021-00825-x.

Hausfather, Z., K. Marvel, G. A. Schmidt, J. W. Nielsen-Gammon, and M. Zelinka, 2022: Climate simulations: recognize the 'hot model' problem. *Nature*, **605**, 26–29, https://doi.org/10.1038/d41586-022-01192-2.

Heede, U. K., A. V. Fedorov, and N. J. Burls, 2020: Time Scales and Mechanisms for the Tropical Pacific Response to Global Warming: A Tug of War between the Ocean Thermostat and Weaker Walker. https://doi.org/10.1175/JCLI-D-19-0690.1.

Hsiao, W., Y. Hwang, Y. Chen, and S. M. Kang, 2022: The Role of Clouds in Shaping Tropical Pacific Response Pattern to Extratropical Thermal Forcing. *Geophys. Res. Lett.*, **49**, https://doi.org/10.1029/2022GL098023.

Kang, S. M., Y. Yu, C. Deser, X. Zhang, I.-S. Kang, S.-S. Lee, K. B. Rodgers, and P. Ceppi, 2023: Global impacts of recent Southern Ocean cooling. *Proc. Natl. Acad. Sci.*, **120**, e2300881120, https://doi.org/10.1073/pnas.2300881120.

Kay, J. E., Y.-C. Liang, S.-N. Zhou, and N. Maher, 2024: Sea ice feedbacks cause more greenhouse cooling than greenhouse warming at high northern latitudes on multi-century timescales. *Environ. Res. Clim.*, **3**, 041003, https://doi.org/10.1088/2752-5295/ad8026.

Myers, T. A., C. R. Mechoso, and M. J. DeFlorio, 2017: Importance of positive cloud feedback for tropical Atlantic interhemispheric climate variability. *Clim. Dyn.*, https://doi.org/10.1007/s00382-017-3978-1.

Rugenstein, M., and Coauthors, 2019: LongRunMIP: Motivation and Design for a Large Collection of Millennial-Length AOGCM Simulations. https://doi.org/10.1175/BAMS-D-19-0068.1.

——, S. Dhame, D. Olonscheck, R. J. Wills, M. Watanabe, and R. Seager, 2023: Connecting the SST Pattern Problem and the Hot Model Problem. *Geophys. Res. Lett.*, **50**, e2023GL105488, https://doi.org/10.1029/2023GL105488.

Rugenstein, M. A. A., and K. C. Armour, 2021: Three Flavors of Radiative Feedbacks and Their Implications for Estimating Equilibrium Climate Sensitivity. *Geophys. Res. Lett.*, **48**, e2021GL092983, https://doi.org/10.1029/2021GL092983.

Webb, M. J., and Coauthors, 2017: The Cloud Feedback Model Intercomparison Project (CFMIP) contribution to CMIP6. *Geosci. Model Dev.*, **10**, 359–384, https://doi.org/10.5194/gmd-10-359-2017.

Zhou, C., M. D. Zelinka, and S. A. Klein, 2016: Impact of decadal cloud variations on the Earth's energy budget. *Nat. Geosci.*, **9**, 871-+, https://doi.org/10.1038/ngeo2828.

---

## Community Comment (CC2)

**Comment on Dunne et al 2025: 'An evolving Coupled Model Intercomparison Project phase 7 (CMIP7) and Fast Track in support of future climate assessment'**

This is a very timely and important paper that lays out the evolution of the CMIP project and details the plans for its next phase, CMIP7. I have a couple of comments;

1. An important part of the design of CMIP7 that differs from earlier phases is the separation of policy relevant simulations (the Fast-Track) from the research orientated simulations designed to address the scientific questions and provide a rich characterisation of climate model capability to support future development. I feel the thinking behind this new development could be made more explicit. In particular, how this came about - at least in part- from the feedback from modelling groups about the burden of CMIP6 simulations. Engagement and support for modelling groups contributing to CMIP7 will be critical and documenting more clearly the influence they had on the design of CMIP7 will give reassurance to the community that they can achieve a balance between delivering to their national agendas as well as engagement in international community science.

2. There are numerous references to the critical role that CMIP has played in underpinning the IPCC assessments. This is absolutely right and an important point to be made. I also think the authors have tried to carefully lay out that CMIP has - and will continue to support the national and international science communities. However what is perhaps missing is a third important role that the policy relevant simulations have played in supporting the national assessments of many countries. A quick question to ChatGPT(!) gives the following 12 countries/communities that have used CMIP scenarios to deliver their national assessments. It would be good to document this important role emphasising the support CMIP plays for national agendas.

a. United States
   • National Climate Assessment (NCA)
   • Led by the U.S. Global Change Research Program (USGCRP)
   • Uses CMIP5 and CMIP6 projections for national and regional climate impact assessments.
   • Latest report: Fifth National Climate Assessment (NCA5, 2023)

b. United Kingdom
   • UK Climate Projections (UKCP)
   • Developed by the Met Office Hadley Centre
   • Uses CMIP5 (UKCP18) and CMIP6 (UKCPNext) to provide probabilistic and high-resolution UK-specific projections.

c. European Union
   • European Climate Risk Assessment (EUCRA)
   • Managed by Copernicus Climate Change Service (C3S) and European Environment Agency (EEA)
   • Uses CMIP6 projections within EURO-CORDEX for downscaled regional assessments.

d. Canada
   • Canada's Climate Change Report (CCCR)
   • Produced by Environment and Climate Change Canada (ECCC)
   • Uses CMIP5 and CMIP6 for projections at the national level.

e. Australia
- State of the Climate Report (by CSIRO & Bureau of Meteorology)
- Climate Change in Australia Projections
- Uses CMIP5 and CMIP6, downscaled for Australian conditions.

f. Germany
- GERICS Climate Fact Sheets (by the Climate Service Center Germany)
- German Climate Change Assessment Report
- Uses CMIP6 projections, often combined with EURO-CORDEX downscaling.

g. France
- Drias Future Climate Scenarios (by Météo-France)
- GREC (Regional Climate Group) Reports
- Uses CMIP5 and CMIP6, combined with CNRM-CM models and EURO-CORDEX.

h. China
- China's Third National Climate Change Assessment Report
- Uses CMIP5 and CMIP6 within China's regional modeling framework (BNU-ESM, FGOALS).

i. Japan
- Climate Change in Japan Report (by Japan Meteorological Agency, JMA)
- Uses CMIP6 and the JRA-55 reanalysis dataset.

j. New Zealand
- NIWA Climate Change Projections
- Uses CMIP5 and CMIP6, often with regional downscaling via VCSN (Virtual Climate Station Network).

k. South Africa
- South African Risk and Vulnerability Atlas (SARVA)
- Uses CMIP5 and CORDEX-Africa for regional climate projections.

---

## Author Response (AR1)

*Reviewer comments in italics,* **Response to reviewers in Bold**

*RC1: Anonymous:*

*CMIP has been a cornerstone of international Earth system modelling for the past 3 decades, delivering key science support to IPCC Assessments, while advancing the development of Earth system and climate models and their use to understand past and future evolution of the climate system. CMIP has been grappling with the dual demands of delivering science support (mainly future projections) to international climate change assessments, and the growing climate service sector, while also coordinating research-led experiments to advance Earth system models and scientific understanding. This dual set of demands has caused CMIP to grow significantly over its past two iterations (CMIP5 and CMIP6) in terms of MIPs, experiments to be run, and data to be archived, with consequences for contributing modelling groups. All of this has (and is) being done through short-term, uncoordinated (at an international level) funding, supporting the development of forcing data, realization of experiments, and maintenance of the underpinning infrastructure. Such a situation is difficult to maintain, something had to change in the organization of CMIP going forwards. CMIP7, as described in this paper, is a step towards such a change, with a first attempt to separate simulations intended to support international assessments (e.g. IPCC AR7 and the CMIP7 Fast Track) from other experiments intended to advance the science and modelling of the climate system (e.g. CMIP7 community MIPs). This paper is therefore timely and important. From this perspective the paper clearly needs to be published, though not in its present form. Below I outline a number of points that need addressing before the paper is suitable for publication. I hope this will increase its value for the research community and for CMIP more generally.*

*Major points.*

*The paper is very wordy, with lots of long sentences and lists justifying why things have been, or will be, done in a certain way. This makes the paper tedious to read. Addressing this could reduce the length of the paper (easily) by 25% and make it a more enjoyable read! As an example, lines 73 to 122 could be reduced to ~10 lines and still deliver the key messages.*

**Worth working on**

*Section 3.5 adds very little.*

**Perhaps it can be shortened further? While it can be argued that the result was mostly a null and so not particularly exciting, the process and conclusions came from a comprehensive evaluation driven by strong community requests to reduce the burden on modeling centers and users and would want to circumvent continued argument that such an effort should again be performed.**

*While the CMIP IPO is a very good development and is doing a great job supporting the development of CMIP7, I am not convinced much of section 4.1 is really needed in a paper.*

**Perhaps it can be shortened further? I think this section is important as an independent scientific justification for continuity of international funding for CMIP**

*Section 4.2 is also very wordy and rambling. This is true for a lot of the introduction, which could be reduced in length without losing much.*

**Perhaps it can be shortened further?**

*There are quite a few examples of repetition. e.g. lines 84-86, lines 93-94, lines135 to 138, 144-145, 440-445. This needs to be reduced throughout the manuscript.*

**Worth looking into.**

*There are also numerous examples of sentences beginning with long justifications for what is to come based on what has already been said: e.g. Line 93: "In addition to the systematic characterization of climate mechanisms…." or line 110: "Beyond direct contribution to national and international climate assessments…and lots of similar examples. I don't think these are needed and can be deleted in lots of places.*

**Worth looking into.**

*The paper has lots of examples explaining how CMIP has (and will) be supported by, aligned with, and deliver to, WCRP. While CMIP is a WCRP-sponsored activity and this is important, it is likely sufficient to say this once (most people know this already) and not have numerous motivations and links to WCRP listed. I suggest reducing these (examples include lines 60-63, 110-120, and others)*

**Worth looking into.**

*The 4 research questions are all interesting, and important, What the paper lacks is a clear link between these research questions and the experiments proposed (either as part of the Fast Track or within the community MIPs). Will there be new experiments designed to specifically address some of the research questions? How will the existing experiments advance understanding? In some cases this is clear (e.g. CO2-emission driven models will likely expose (and lead to improvement in) carbon-cycle biases and feedbacks more thoroughly than concentration-driven models) but in many instances it isn't. The connection between the guiding research themes and the experiments planned in CMIP7 needs to be better explained.*

**Perhaps Robert Pincus and Tatiana Ilyina can take this on?**

*In two places (line 180 and line 645) there is an assertion high ECS models in CMIP6 have been proven to be incorrect and by implication these models are worse than lower ECS models, or just wrong. I don't agree with this assertion. A high ECS Earth (>5K) is very unlikely, but it has not been conclusively ruled out. If anything, recent increased warming and suggestions of a possible role for changing cloud-radiation processes in this increased warming, may increase the likelihood of a high ECS world. In addition, some of the CMIP6 models with high (increased relative to CMIP5) ECS have been shown to realize this because of improvements in specific cloud feedback processes that were previously (erroneously) balancing other incorrect feedback magnitudes leading to a lower ECS through compensating errors. With removal (improvement) in one aspect of this compensation, ECS has increased. While the higher ECS may not be correct, the underlying processes/feedbacks are likely*

*simulated better. To me this is a model improvement. It would be a pity if CMIP7 discouraged groups from making such important model improvements, even if that risked increasing their model ECS value. I suggest modifying these two assertions.*

**Fair point from a model development perspective – worth making sure to limit the assertion to the emergent behavior of the models being inconsistent with historical warming and thus less useful from a climate services perspective.**

*The general aspiration for CMIP7 to separate out policy-relevant simulations (e.g. Fast Track for IPCC AR7) and longer-term MIPs aimed at specific research questions, is a good one. The paper could do a better job explaining and motivating this separation, including how modelling groups could best contribute to either or both parts of CMIP7.*

**Thoughts on how to do this?**

*Table 3 is very long and poorly explained. Could it be presented in a more engaging manner? If the main explanations for the different experiments are in the references listed in the table, please let the reader know that. Also, I think there may be some errors in the table. e.g. (i) are piClim-histaer and piClim-histall 30y AMIP or 172y AMIP runs? (ii) For piClim-X and SSPXSST-SLCF I don't see how feedbacks can be assessed (as suggested in the table) if the models are run in prescribed-SST mode. At least the classical definition of a feedback modifying the SST-response to a given forcing and thus also modifying the forcing itself, cannot be realised in prescribed-SST mode. (iii) for piClimSLCF it is unclear what happens to the non-SLCF emissions. Are these held at PI values? A bit better explanation of this table would help the reader.*

**Does satisfying the first request just mean adding "Main explanations for the different experiments are in the references listed in the table " to the table caption?  Need to go through table with the RFMIP/CFMIP experts**

*Minor Points*

*On the "guiding research questions" I don't understand why these are "ephemeral" (line 155).*

**Probably best to eliminate that sentence and add "expanded simulations with coupled carbon-climate ESMs" to the next**

*Regarding the Fast Track experiments, it is not clear if groups are recommended to do everything in either emission-mode or concentration-mode. For example, are there plans for DAMIP to support both emission-driven and concentration-driven experiments? This is not made clear in the explanation of table 3.*

**Agreed, We need to add a statement that we recommend running DAMIP experiments in CO2 concentration mode.**

*Line 494-495: How will DAMIP support analysis of individual forcings in the context of an interactive carbon cycle? Will DAMIP run a coordinated set of experiments for emission-driven ESMs?*

**Beyond stating the above, any DAMIP community MIP is outside the scope of this paper.**

*Line 128 talks about the lack of infrastructure for a sustained approach. This is also true with respect to funding of modelling groups to realize such regular simulations. This should also be highlighted.*

**Agreed.**

*Lines 221-223 on high resolutions models contradicts itself. Please make clearer what you mean here.*

**Need to remove the word "modest"**

*In section 2.3 I am surprised that emission-driven ESM (scenarioMIP) projections are not discussed more. This seems an important development on CMIP6.*

**Can Tatiana Ilyina help here?**

*Lines 266 to 267: while modelling groups suggest that increase in fire over this century (Allen et al. 2024) seems to be an incomplete sentence.*

**Need to add comma before "while and replace "suggests that" with "projects a large"**

*For section 2.4 more discussion on potential MIP contributions to addressing this seems appropriate (e.g. TIPMIP, CDRMIP, C4MIP). I am also surprised there isn't more mention of global warming overshoot scenarios in this section.*

**Agreed, need to add**

*Line 350: coupled carbon-climate ESMs importance in climate stabilization is mentioned. The importance in negative emission scenarios (warming overshoot) is likely even more important to mention.*

**Agreed, need to add**

*Lines 386-388: Will there be a coordinated effort to compare CMIP6 historical and scenario forcings to those in CMIP7? This would be a good thing to do (e.g. a forcing comparison MIP).*

**Can we just cite the CMIP7 forcings special issue.**

*Section 3.2. Will there be any stability/conservation requirements to meet for the piControl or esm_piControl runs?*

**Need to add that we are maintaining the CMIP6 C4MIP criterion of 10 PgC/century per component.**

*Lines 421 to 425: I don't understand what is being proposed here. Please make it clearer.*

*If model X is used in a given science MIP, is it still an entry-card that model-X also runs the CMIP7 DECK? This is not clear.*

*Line 628: The REF is mentioned and somewhere else this is defined as a Rapid Evaluation Framework. What the REF is, and what it is intended for, needs to be more clearly explained.*

**Can Birgit Hassler help with this?**

*RC2: Chris Jones:*

*Review of CMIP7 documentation paper, by Dunne et al. Firstly to say that the CMIP panel and authors here are to be congratulated on the way they have approached the task of developing CMIP7 plans in a complex landscape of requirements. CMIP has had a lot of success historically but requirements have grown and that growth is not sustainable so the new approach to consult with both users and providers and hence prioritise a more manageable, but still vital, set of simulations has been extremely welcome. The outreach, consultation and dissemination of information has been excellent throughout and this paper contributes to that process. CMIP is a huge undertaking and changes the deployment of resource (both personal and computing/technology) in many, many modelling and research centres around the world. Careful design of what is requested and why is essential. I perform this review mainly in the context that the main aspects of CMIP7 and the Fasttrack, are already determined and too late to make substantial changes. Therefore, I focus on the presentation and explanation aspects with a few suggestions of things which could still be tweaked or clarified. My major comment is to ask for more details on where the "Guiding Research Questions" came from? Are these the result of a consultation on the priority climate science questions? They resemble, but are not the same as, past WCRP grand challenges (e.g. on extremes or carbon cycle).*

**These questions were certainly consistent with the WCRP Grand Challenges but nurtured somewhat organically for within the CMIP Panel after the Fast Track was established but largely independent from it in an attempt to highlight the key emergent themes/needs for new modeling – to answer the question of Why is a new CMIP necessary?... At the time we considered conducting a formal survey, but then decided we needed to capture a more in-depth perspective motivating the CMIP Panel. I will have to think more about the best way to frame this.**

*The way the paper is presented implies you started with these as a guiding set of questions and designed CMIP7 to answer them.*

**No, the questions came after the fast track experiments were defined. The fast track experiments were largely based on the subset of CMIP6 experiments that were most highlighted in AR6. The questions, rather, were driven by the remaining gaps post AR6.**

*But in practice that wasn't how I recall it happening – so have these questions been retro-fitted to the experiments? E.g. line 132 says that CMIP7 design came from consultation and surveys – this is certainly true of the experiments – but did this consultation also take place for the science questions?*

**No, the science questions came from the expert judgement from within the CMIP Panel.**

*When I look over the CMIP7 web page there are lots of details and further links to the experiments, the task teams, the data request, the REF etc. Your figure 3 is replicated on the website, which mentions the science questions linked to each FT experiment - but I cannot see the questions described or explained anywhere. It feels like these questions have been added after the experiment design. If these really are "guiding questions" that have guided, and are intended to keep guiding, CMIP I think they and their derivation need more prominence. It is not clear, for example, why you identify SST patterns over, say, cloud feedbacks, as a key driver of system sensitivity?*

**We should clarify that the SST patterns are an example of a post-CMIP6/AR6 challenge with an observational constraint that needs to be causally explained that may or may not be related to climate sensitivity.**

*Also, when you discuss a "carbon-water nexus" – is this just a catch-all for things not included in the other questions? The paragraphs of description of this question (sec 2.3) don't appear to cover interactions between carbon and water cycles as implied by the "nexus" tag.*

**Very fair criticism – we should do a better job at explaining that the reason for calling it a "water-carbon nexus" is that the CMIP Panel sees water and land carbon as the scope of a set of fundamentally linked problems.**

*So overall it would be good to articulate maybe how these priorities were arrived at. I am not querying the importance of these questions – they are clearly crucial. But other aspects (for example on aerosol forcing and cloud processes) could also be seen as equally important, and CMIP7 will address many more than just these. Maybe it is better to present the experiments first and then give some example high priority questions as examples of things which CMIP7 may help address – but it feels to be overselling the tag of "guiding questions" to imply that these came first and led to the CMIP7 design.*

**The re-ordering of the Science Questions first was driven by Robert Pincus – hopefully, he can lead the justification and edits here.**

*Other suggestions I think are important: Model/simulation quality i. ii. Lines 374-375 – it feels reasonable to suggest a degree of stability of a control run: +5ppm is probably OK – but better as a rate than an absolute – is this +-5ppm per century for example? In CMIP6 C4MIP requested drifts of less than 10 PgC per century in the main pools.*

**Agreed, as requested by RC1.**

*But it would be consistent to also request stability criteria for other metrics – e.g. global T must drift by no more than +- XX degrees, or AMOC within XX Sv. It would be good to treat all major climate components similarly.*

**This was something Helene and I attempted but received a lot of push-back. Worth discussing again – historical mean SST bias < X?, AMOC greater than 10 Sv?, interannual SST variability >Y (i.e. ENSO/AMO, PDO, SAM)?**

*More importantly – I think it is unwise, however, to suggest arbitrary quality criteria for historical runs. Many ESMs may not hit the historical CO2 within 5ppm. See e.g. Hajima et al (https://egusphere.copernicus.org/preprints/2024/egusphere-2024-188/) for thorough evaluation of CMIP6 models in this respect. What happens if a model does not hit you 5ppm bounds – is it excluded from analysis?*

**We need to look back at the C4MIP guidance on "stable" PI Control carbon budget, but I think it was just a target, not a "requirement" to put data on ESGF. Users will decide whether to use the output in their analysis.**

*Again – as above, will you also specify acceptance criteria on other measures? – e.g. goodness of fit of the historical temperature record?*

**No, I don't think any of the "goodness of fit" is about requirements for ESGF, it is just guidance based on "fitness for purpose" at representing historical climate.**

*This would be a big change for CMIP – to specify acceptance criteria – I think it needs much more consultation before you introduce this.*

**The consultation process was the SED TT – we should go back to the SED TT co-authors to formulate our response.**

*Ensembles – do you have any recommendations around generation of ensembles (from each model)? I realise you don't want to rule out models by requiring large ensembles, but some experiments may benefit more than others from ensembles.*

**Thoughts on what this guidance should be and where it should go?  I think SEDTT had guidance on this at some point – Isla Simpson and Ben Sanderson?**

*Line 510 says that the FT "promotes the generation of ensembles" – but it is not clear how? FT does not appear to mention ensembles at all – but it could be a good opportunity to do so. It might be useful to provide guidance on this without mandating.*

**The current language was referring to the CMIP ensemble, not the ensembling of a single model, but the point is well-taken,**

*Likewise you could guide on choice of initial conditions (e.g. branch points best taken >XX years apart from the control run).*

**Thoughts from Isla Simpson and Ben Sanderson?**

*As an example, quantifying TCRE from flat10 is a relatively large signal-to-noise activity. Ensembles may add little value to this. But quantifying ZEC from the flat10-zec simulation is a very small signalto-noise and ensembles of this run could be really useful. See e.g. Borowiak et al (https://agupubs.onlinelibrary.wiley.com/doi/full/10.1029/2024GL108654) which shows that ZEC derived from CMIP6 ZECMIP are subject to a level of uncertainty which CMIP6 did not consider due to lack of ensembles.*

**Clarificattion that we are refering to multi-model ensembles to assess structural variability rather than sing-model ensembles to assess internal variability should resolve this**

*Spin-up. I'm not sure I understand the request to submit numerical results from the spin-up of the models. What is the goal of this – how will they be used? "for curation" sounds like an odd phrase – why do these need curating? And what does "curation" involve – is this the same as archiving on a public database like ESGF?*

**Agreed that we need to provide a better justification than "curation" – indeed, our hope was that a "Fresh Eyes" team would perform an analysis of this dataset and that it would be in general useful information to researchers doing analysis on the potential role of spinup as a form of "structural uncertainty" and "internal variability"**

*Model selection. I think you are very wise not to do any prior screening or selection of models. The "hot models" paper you cite in Appendix 3 by Hausfather et al is rather simplistic to provide a table of "Y" and "N" on model screening based on sensitivity. A more nuanced analysis by Swaminathan et al (https://agupubs.onlinelibrary.wiley.com/doi/full/10.1029/2024EF004901) shows clearly that many metrics of crucial interest are not related to ECS. Many high sensitivity models have very good evaluation*

*scores on many metrics and vice versa – having a lower ECS is certainly not a measure of quality. Any screening or selection needs to be much better understood and carried out case-by-case for the application in question. It cannot (yet) be done at the scale of CMIP which has so many downstream uses of the outputs.*

**We need to find a way to incorporate these points into the model selection section.**

*Minor comments*

*• Lines 102-107. This is a nice description of how CMIP has expanded and refined focus as both the expertise and need evolves. It feels that more knowledge of reversibility and symmetry is a big gap in our understanding of the climate system, and here could be a good place to articulate the need for more process exploration of how the system behaves under reversing of forcing.*

*• Line 216 says that CMIP7 focus on emissions-driven runs allows for more exploration of extremes under stabilisation – can you explain how so?*

*• Sec 2.4 on points of no return – is there a reason not to call this either "tipping points" or "irreversibility" which have become much more common phrases for these topics. Wood et al (2023 - https://agupubs.onlinelibrary.wiley.com/doi/full/10.1029/2022EF003369) is a good reference here for the framing of high impact/low likelihood outcomes and the need for research spanning different dimensions of this topic.*

*• Line 297 onwards – describing the CMIP7 DECK intent. It is worth being explicit here that the goal is only to characterise the response to _increasing_ forcing. It was a deliberate decision not to add a DECK experiment to characterise the system response to reducing forcing. (This remains a gap in CMIP7 – noting that flat10-cdr can only be performed by ESMs)*

*• Table 1 is important. A couple of notes/suggestions - For esm-piControl the forcing is described as "emissions" - I wonder if this should be better described as "interactive $CO_2$" or "simulated $CO_2$" because of course there are no emissions. So even though we informally describe this as "emissions mode" it risks implying that there are some emissions being applied. Or at least specify that $CO_2$ emissions are zero. - Typo – looks like the 1% and historical lines have transposed the solar/volcanic forcing entries • Line 355. Can you clarify the need for 100 years of control run before any experiments are branched off? I don't recall this being requested in CMIP6*

*• Line 364 – can you explain why conc-driven control run is required if the esm-control is stable? That seems redundant*

*• Table 2 is useful – but it feels odd to name individuals. What happens as/when a person moves job etc? maybe a named group in an organisation is more useful.*

*• Table 2, N deposition. Will this be speciated into dry/wet and oxidised/reduced reactive nitrogen?*

*• Line 405. The section on spin-up – it is not clear how the strap line "characterising model diversity" is relevant to this sub-section. Maybe just call the section "ocean and land spin-up" (where land here includes land ice/cryosphere?)*

*• Line 470 – is "SCP" a typo? "SSP"?*

• Table 3 is super useful and important – it will be a very good easy-look-up of the whole set of FT simulations. But it is really big! It is important that it is produced and typeset to be easily readable given how big it is. I feel this comment may be more for the journal/typesetters than the authors – I hope you can find a way to make it well readable.

• Table 3 – scenario time period. You quote that scenarios run to 2100 – is this decided? I thought it would be 2125, or at least this was still being discussed. (personal opinion – it drives me mad that IPCC figures and values can only ever quote a climate – i.e. 20-year average – for 2090. So an extension to a minimum of 2110 seems vital so that we can actually quote a 2100 value for projected results!)

• Appendix 1 – requested spin-up metrics. As per my comment above I'm not yet convinced why you need to request these. But if you do, then to close the land carbon cycle you should also requested cProduct. Even if the control run has no land-use _change_ it will still have land use, and the product pools may well be non-zero. cLand is then the sum of cVeg+cLitter+cSoil+cProduct

*CC1, Mark Zelinka:*

*Review of "An evolving Coupled Model Intercomparison Project phase 7 (CMIP7) and Fast Track in support of future climate assessment" by Dunne et al [egusphere-2024-3874]*

*Summary The authors motivate and describe the seventh iteration of CMIP, including the new Fast Track set of experiments which serves the IPCC. The paper is mostly effective in achieving these goals, but there are a few areas needing improvement. This review largely deals with issues relevant to the Cloud Feedback Model Intercomparison Project (CFMIP). Mark Zelinka Maria Rugenstein Alejandro Bodas-Salcedo Jennifer Kay Paulo Ceppi Mark Webb on behalf of the CFMIP Scientific Steering Committee*

*Major Comments*

*● Section 2.1 describes the first of four guiding questions in CMIP7, dealing with pattern effects. A large part of the reason the scientific community is interested in pattern effects is because of the science conducted by members of the CFMIP community (Andrews et al. 2015; Zhou et al. 2016; Andrews and Webb 2017; Ceppi and Gregory 2017; Andrews et al. 2018, 2022), facilitated by CFMIP experiments like amip-piForcing (Andrews 2014; Webb et al. 2017), and illuminated by CFMIP diagnostics (including satellite cloud simulator diagnostics that reveal the diverse cloud responses to warming patterns). The "Why expect progress now?" section completely excludes a role for CFMIP while instead mentioning the roles that can be played by DAMIP and AerChemMIP. The focus here seems to be more on what causes warming patterns (a worthy goal), but the understanding of the climate response (including but not limited to clouds) to diverse warming patterns is essential to this problem and should not be neglected. Moreover, the surface temperature response pattern is likely to be at least partly affected by how clouds and their radiative effects feed back on warming patterns (Myers et al. 2017; Erfani and Burls 2019; Rugenstein et al. 2023; Espinosa and Zelinka 2024; Breul et al. 2025) and are involved in teleconnections that propagate surface temperature anomalies from high to low latitudes (Kang et al. 2023; Hsiao et al. 2022). We suggest better acknowledging CFMIP contributions to the current understanding of the pattern effect and explicitly calling out the role that CFMIP can play in making progress. We also note that the first sentence of this paragraph is rather hard to parse and is formulated rather weakly ("xyz may all help" – it remains unclear with what and how).*

● CFMIP requests that the abrupt CO2 experiments (4x, 2x, and 0.5x) be run out to a minimum of 300 years, and we strongly encourage modeling groups to run beyond that (which could be noted at L331). Note that CFMIP requested this minimum duration as part of the FastTrack consultation process, which was then adapted into the request for the abrupt CO2 experiments. (See the abrupt-4xCO2 request: https://airtable.com/embed/appVPW6XAZfbOZjYM/shrqq9I4NJThwOT9W/tblkc1IkKEtiY Kcho/viw9PLlrOnfUMcvHw/recl01t59HM8jz8ax.) Table 1 currently lists the abrupt-4xCO2 run as extending for "150+ (300)", though it is not clear what this nomenclature means exactly.  We request that "150+" be replaced with "300+" to make it clear that 300 years is the desired minimum, and "(300)" be replaced with "(1000)". The reasons for requesting that the abrupt CO2 runs be integrated for a minimum of 300 years with strong encouragement to extend beyond that are manifold:  ○ Better ECS quantification: Rugenstein and Armour (2021) quantified with 10 equilibrated CMIP5 and CMIP6 models that 400 years are necessary to estimate the true equilibrium climate sensitivity within 5% error. The model spread in equilibration is large and CMIP6/7 models probably need longer to equilibrate due to the "hot model problem" (Hausfather et al. 2022), which partly consists of temperature- and time-dependent feedbacks. Kay et al (2024) estimated an equilibrium timescale of 200+ years for 2xCO2 and 500+ years for 0.5xCO2, noting important implications for paleo cold climate constraints (e.g., LGM) that can only be understood if the simulations are long enough. ○ Understanding centennial coupled behavior: Simulations of at least 300 years are necessary for estimating the pattern effect, ocean heat uptake and convection (Gjermundsen et al. 2021), AMOC recovery (Bonan et al. 2022), and Equatorial Pacific response timescales (Heede et al. 2020). ○ Understanding and quantifying feedback temperature dependence: This is not well understood, could lead to tipping points and is, after the pattern effect and cloud feedbacks, the biggest unknown in estimating ECS, understanding hot models, and high-risk futures (Bloch-Johnson et al. 2021). It is very hard to quantify because it is obscured by the pattern effect, but is aided by longer simulations.  ○ Practical considerations: Running existing simulations for longer is typically easier than running new simulations. Thus, if computing time is available at modeling centers, it is strongly encouraged that pre-industrial control and abrupt CO2 runs be extended as long as possible. Anecdotally, many of the model centers contributing to LongRunMIP (Rugenstein et al. 2019) had independently run their simulations for longer than 150 years and had the data sitting around, suggesting that in many cases such long simulations are already being performed or are trivial to extend. Currently, ~52 groups are using the LongRunMIP simulations for studies on internal variability, global warming levels, feedback quantification, paleo climate, oceanography, and training for data-driven machine learning approaches. Minor Comments

● L34: Should it be "...include experiments to diagnose historical..."?

● Introduction section: This section may be too long. The main audience of this paper is the science community that want to understand the rationale and details of the experimental design, not the history of CMIP iterations.

● L90: should be Zelinka et al 2020

● L125-127: Suggest being more specific and use "modeling community", rather than "research community" as a whole. The research community benefits as a whole, but it doesn't share the burden.

● L130" "... the present experimental design includes some components …" This point is hard to parse. The entire paragraph reads well though, but the role DECK plays in climate services might need more

*highlighting. The remainder of the paper is phrased mostly in terms of science questions and the role climate service plays in there remains somewhat unclear.*

● *L140: Would it be worth listing a few big questions which were answered mainly or only through past CMIP cycles?*

● *L265-266: something wrong with the phrasing here*

● *Table 1: It's unclear why the request is for a small ensemble for historical and a large ensemble for amip*

● *Section 3.1.2: It would be helpful to see a plot of how the new forcing datasets differ from those used in CMIP6 during the 1850-2014 period.*

● *L310/Fig.2: This schematic might benefit from a vertical time axis. The current version leaves a lot of room for interpretation. What are the small orange arrows? What is the connection between DECK and AR7 Fast Track?*

● *L355: "year 100 or later of piControl" – is the rationale for this given anywhere in the manuscript?*

● *L383: The historical and AMIP simulations end in 2021 according to Table 1.*

● *L498: CFMIP deals with cloud and non-cloud feedbacks (all radiative feedbacks)*

● *L501: Figure 3 excludes RFMIP from the "Characterization" box, yet it is highlighted in this Characterization section, which is confusing.*

● *L510-511: Very hard to parse this statement*

● *L516: "Forcing" should be "Feedback"*

● *L517: I believe you mean "CFMIP" rather than (or in addition to) "CMIP" here*

● *L541: Missing section number*

● *Table 3, amip-p4K: missing word here? "feedbacks observed"*

● *Table 3, amip-p4K: the number of years should be 44 (1979 - 2022)*

● *Table 3, amip-piForcing: the number of years should be 153 (1870 - 2022)*

● *L638: 4 should be 3*

● *Appendix 1 table: Suggest specifying top of atmosphere albedo when referencing rsdt and rsut*

● *L712-713: Might be some missing words here References Andrews, T., 2014: Using an AGCM to Diagnose Historical Effective Radiative Forcing and Mechanisms of Recent Decadal Climate Change. https://doi.org/10.1175/JCLI-D-13-00336.1. ——, and M. J. Webb, 2017: The Dependence of Global Cloud and Lapse Rate Feedbacks on the Spatial Structure of Tropical Pacific Warming. J. Clim., 31, 641–654, https://doi.org/10.1175/JCLI-D-17-0087.1. ——, J. M. Gregory, and M. J. Webb, 2015: The Dependence of Radiative Forcing and Feedback on Evolving Patterns of Surface Temperature Change in Climate Models. J. Clim., 28, 1630–1648, https://doi.org/10.1175/jcli-d-14-00545.1. ——, and Coauthors, 2018: Accounting for Changing Temperature Patterns Increases Historical Estimates of Climate*

*Sensitivity. Geophys. Res. Lett., 0, https://doi.org/10.1029/2018GL078887. ——, and Coauthors, 2022: On the Effect of Historical SST Patterns on Radiative Feedback. J. Geophys. Res. Atmospheres, 127, e2022JD036675, https://doi.org/10.1029/2022JD036675. Bloch-Johnson, J., M. Rugenstein, M. B. Stolpe, T. Rohrschneider, Y. Zheng, and J. M. Gregory, 2021: Climate Sensitivity Increases Under Higher $CO_2$ Levels Due to Feedback Temperature Dependence. Geophys. Res. Lett., 48, e2020GL089074, https://doi.org/10.1029/2020GL089074. Bonan, D. B., A. F. Thompson, E. R. Newsom, S. Sun, and M. Rugenstein, 2022: Transient and Equilibrium Responses of the Atlantic Overturning Circulation to Warming in Coupled Climate Models: The Role of Temperature and Salinity. https://doi.org/10.1175/JCLI-D-21-0912.1. Breul, P., P. Ceppi, and P. Nowack, 2025: The importance of stratocumulus clouds for projected warming patterns and circulation changes. EGUsphere, 1–22, https://doi.org/10.5194/egusphere-2025-221. Ceppi, P., and J. M. Gregory, 2017: Relationship of tropospheric stability to climate sensitivity and Earth's observed radiation budget. Proc. Natl. Acad. Sci. U. S. A., 114, 13126–13131, https://doi.org/10.1073/pnas.1714308114. Erfani, E., and N. J. Burls, 2019: The Strength of Low-Cloud Feedbacks and Tropical Climate: A CESM Sensitivity Study. J. Clim., 32, 2497–2516, https://doi.org/10.1175/jcli-d-18-0551.1. Espinosa, Z. I., and M. D. Zelinka, 2024: The Shortwave Cloud-SST Feedback Amplifies Multi-Decadal Pacific Sea Surface Temperature Trends: Implications for Observed Cooling. Geophys. Res. Lett., 51, e2024GL111039, https://doi.org/10.1029/2024GL111039. Gjermundsen, A., A. Nummelin, D. Olivié, M. Bentsen, Ø. Seland, and M. Schulz, 2021: Shutdown of Southern Ocean convection controls long-term greenhouse gas-induced warming. Nat. Geosci., 14, 724–731, https://doi.org/10.1038/s41561-021-00825-x. Hausfather, Z., K. Marvel, G. A. Schmidt, J. W. Nielsen-Gammon, and M. Zelinka, 2022: Climate simulations: recognize the 'hot model' problem. Nature, 605, 26–29, https://doi.org/10.1038/d41586-022-01192-2. Heede, U. K., A. V. Fedorov, and N. J. Burls, 2020: Time Scales and Mechanisms for the Tropical Pacific Response to Global Warming: A Tug of War between the Ocean Thermostat and Weaker Walker. https://doi.org/10.1175/JCLI-D-19-0690.1. Hsiao, W., Y. Hwang, Y. Chen, and S. M. Kang, 2022: The Role of Clouds in Shaping Tropical Pacific Response Pattern to Extratropical Thermal Forcing. Geophys. Res. Lett., 49, https://doi.org/10.1029/2022GL098023. Kang, S. M., Y. Yu, C. Deser, X. Zhang, I.-S. Kang, S.-S. Lee, K. B. Rodgers, and P. Ceppi, 2023: Global impacts of recent Southern Ocean cooling. Proc. Natl. Acad. Sci., 120, e2300881120, https://doi.org/10.1073/pnas.2300881120. Kay, J. E., Y.-C. Liang, S.-N. Zhou, and N. Maher, 2024: Sea ice feedbacks cause more greenhouse cooling than greenhouse warming at high northern latitudes on multi-century timescales. Environ. Res. Clim., 3, 041003, https://doi.org/10.1088/2752-5295/ad8026. Myers, T. A., C. R. Mechoso, and M. J. DeFlorio, 2017: Importance of positive cloud feedback for tropical Atlantic interhemispheric climate variability. Clim. Dyn., https://doi.org/10.1007/s00382-017-3978-1. Rugenstein, M., and Coauthors, 2019: LongRunMIP: Motivation and Design for a Large Collection of Millennial-Length AOGCM Simulations. https://doi.org/10.1175/BAMS-D-19-0068.1. ——, S. Dhame, D. Olonscheck, R. J. Wills, M. Watanabe, and R. Seager, 2023: Connecting the SST Pattern Problem and the Hot Model Problem. Geophys. Res. Lett., 50, e2023GL105488, https://doi.org/10.1029/2023GL105488. Rugenstein, M. A. A., and K. C. Armour, 2021: Three Flavors of Radiative Feedbacks and Their Implications for Estimating Equilibrium Climate Sensitivity. Geophys. Res. Lett., 48, e2021GL092983, https://doi.org/10.1029/2021GL092983. Webb, M. J., and Coauthors, 2017: The Cloud Feedback Model Intercomparison Project (CFMIP) contribution to CMIP6. Geosci. Model Dev., 10, 359–384, https://doi.org/10.5194/gmd-10-359-2017. Zhou, C., M. D. Zelinka, and S. A. Klein, 2016: Impact of decadal cloud variations on the Earth's energy budget. Nat. Geosci., 9, 871-+, https://doi.org/10.1038/ngeo2828.*

**It would be great if someone on the author team more familiar with CFMIP could take the lead on this… Robert Pincus?**

*CC2, Cath Senior:*

*Comment on Dunne et al 2025: 'An evolving Coupled Model Intercomparison Project phase 7 (CMIP7) and Fast Track in support of future climate assessment'  This is a very timely and important paper that lays out the evolution of the CMIP project and details the plans for its next phase, CMIP7. I have a couple of comments;*

*1. An important part of the design of CMIP7 that differs from earlier phases is the separation of policy relevant simulations (the Fast-Track) from the research orientated simulations designed to address the scientific questions and provide a rich characterisation of climate model capability to support future development. I feel the thinking behind this new development could be made more explicit. In particular, how this came about - at least in part- from the feedback from modelling groups about the burden of CMIP6 simulations. Engagement and support for modelling groups contributing to CMIP7 will be critical and documenting more clearly the influence they had on the design of CMIP7 will give reassurance to the community that they can achieve a balance between delivering to their national agendas as well as engagement in international community science.*

**We will have to decide the best way to address this.  I think this historical context and justification is one of the disadvantages of the Science Questions coming first.**

*2. There are numerous references to the critical role that CMIP has played in underpinning the IPCC assessments. This is absolutely right and an important point to be made. I also think the authors have tried to carefully lay out that CMIP has - and will continue to support the national and international science communities. However what is perhaps missing is a third important role that the policy relevant simulations have played in supporting the national assessments of many countries. A quick question to ChatGPT(!) gives the following 12 countries/communities that have used CMIP scenarios to deliver their national assessments. It would be good to document this important role emphasising the support CMIP plays for national agendas.  a. United States • National Climate Assessment (NCA) • Led by the U.S. Global Change Research Program (USGCRP) • Uses CMIP5 and CMIP6 projections for national and regional climate impact assessments. • Latest report: Fifth National Climate Assessment (NCA5, 2023) b. United Kingdom • UK Climate Projections (UKCP) • Developed by the Met Office Hadley Centre • Uses CMIP5 (UKCP18) and CMIP6 (UKCPNext) to provide probabilistic and highresolution UK-specific projections. c. European Union • European Climate Risk Assessment (EUCRA) • Managed by Copernicus Climate Change Service (C3S) and European Environment Agency (EEA) • Uses CMIP6 projections within EURO-CORDEX for downscaled regional assessments. d. Canada • Canada's Climate Change Report (CCCR) • Produced by Environment and Climate Change Canada (ECCC) • Uses CMIP5 and CMIP6 for projections at the national level. e. Australia • State of the Climate Report (by CSIRO & Bureau of Meteorology) • Climate Change in Australia Projections • Uses CMIP5 and CMIP6, downscaled for Australian conditions. f. Germany • GERICS Climate Fact Sheets (by the Climate Service Center Germany) • German Climate Change Assessment Report • Uses CMIP6 projections, often combined with EURO-CORDEX downscaling. g. France • Drias Future Climate Scenarios (by Météo-France) • GREC (Regional Climate Group) Reports • Uses CMIP5 and CMIP6, combined with CNRM-CM models and EURO-CORDEX.*

*h. China • China's Third National Climate Change Assessment Report • Uses CMIP5 and CMIP6 within China's regional modeling framework (BNU-ESM, FGOALS). i. Japan • Climate Change in Japan Report (by Japan Meteorological Agency, JMA) • Uses CMIP6 and the JRA-55 reanalysis dataset. j. New Zealand • NIWA Climate Change Projections • Uses CMIP5 and CMIP6, often with regional downscaling via VCSN (Virtual Climate Station Network). k. South Africa • South African Risk and Vulnerability Atlas (SARVA) • Uses CMIP5 and CORDEX-Africa for regional climate projections.*

**This is an excellent point, but I am not sure how to include this information without overwhelming the text… perhaps the IPO can set up a web page on the use of CMIP in National Assessments and we just site that?**

*CC3, Annalisa Cherchi:*

*Broad and comprehensive article to describe forthcoming CMIP7 effort. Some comments below:*

*- among the challenging questions, section 2.3 about the water-carbon-climate nexus does not fully exploit water and the importance of the hydrology processes. We know there are still weaknesses and limitation in this (i.e. Douville et al 2021 in last IPCC AR6 and beyond) but there are now more efforts in modelling centres in this direction;*

**Similar to comment by RC2, we should do a better job at making this link explicit.**

*- Fig 1: the term multiverse seems not fully appropriate as what is shown need and depends on coupling and feedbacks between components and processes. Even here the hydrology part is not fully exploited/described. For example, monsoons are missing among the phenomena. The land interaction is expressed mostly in terms of vegetation and carbon cycle but land is also interaction with the atmosphere via moisture and heat exchanges. In the caption of the figure, red and blue are mentioned as colors for atmosphere and ocean, what about land and cryosphere for example?*

**These points go to the question of the appropriate granularity and comprehensiveness in the Figure… will need to consider.**

*Also related to this, in lines 50-53 model development need to consider and properly represent the coupling between the new components, cryosphere but also improved land-hydrology*

**I am not sure what exactly in land-hydrology needs to be improved – process representation? wetting/drying? Applicability to specific societally relevant questions? Perhaps Isla Simpson can help here?**

*- In term of outline of the paper, the key points highlighted in the abstract (lines 33-38) are not fully exploited within the text, either in terms of sectioning and mostly in the summary.*

**Need to make sure these align**

*In addition the summary (section 5) is not a real summary but mostly contains points of discussion and also new features of this CMIP cycle not described in the sections before, ie. Fresh Eyes on CMIP.*

**Perhaps we should relabel this as a Discussion? Or we can add the Fresh Eyes and over parts back into the earlier sections (They were in previous versions but got removed for length)**

*Also the concept of emulators would deserve a bit more of clarification/explanation. Eventually these new aspects could be more extensively described in this manuscript, leaving some details of the experiments to forthcoming papers.*

**These suggestions would make the manuscript even longer… I am not sure we want to do that.  I am not sure which details of the experiments can be removed.  I worry that many of the details were added because they relate to the application of experiments to multiple questions/MIPs and may not have been explained in the CMIP6 documentation or would lead to dependencies on future publications that may not occur.**

*For example, there are references to details of ScenarioMIP that is not published yet. There is probably no need for those details at this stage as they will described and explained in details once the reference papers will be ready.*

**Now that the ScenarioMIP manuscript has been released, we should go back and much sure this description is in alignment and not redundant or conflicting.**

*A description (outline) of the content of the manuscript could be useful at the end of the Introduction.*

**Would that be helpful, or just make the manuscript even longer?  I think RC1 would disagree with this suggestion.**

*- Overall there are some repetitions (mostly of concept) that could be avoided to simplify the reading (for example, lines 60-76 contains repetitions in the two paragraph and the text could be rewritten and lightened), there are some typos in section 5 (section numbering).*

**Similar to suggestion of RC1.**

---

## Author Response (AR2)

Response (**in bold**) to May 9, 2025 revision reviewer 2 (*in italics*):

*The revised manuscript offers a comprehensive blueprint for CMIP7, detailing the expanded DECK, the Assessment Fast Track (AFT) suite, and the rationale behind the new emissions-driven and process-oriented experiments. Relative to the first revision, the paper is notably clearer, better structured, and more tightly linked to the four flagship science questions. These improvements make the manuscript substantially more readable and impactful. However, the interaction between carbon and water cycles remains insufficiently addressed in the revised Section 2.3. The manuscript also still contains overly long sentences and occasional grammatical errors, which at times impede readability. Further proofreading is recommended. I suggest a minor revision focused on strengthening Section 2.3 and improving the clarity of the writing. A number of specific comments are provided below for the authors' consideration.*

**We thank the reviewer for their attention and commends and have addressed them in the below responses and through another general round of careful reading for language to clarify the points including further clarification of guidance on Scenarios. We have added emphasis on opportunities for coupled water-climate-carbon in the Transient Climate Response to Cumulative emissions and expectations of improved soil carbon representation in Section 2.3.**

*• Lines 64–70: "The historical publicly availability of CMIP ensembles have…" → "has." Also, the sentence is too long. Suggest splitting: "…in house. This accessibility has advanced…"*

**We have fixed the singular verb and broken up the sentence.**

*• Line 95: "Unfortunately. the necessary ESM capabilities…" → "Unfortunately, the necessary…"*

**Fixed**

*• Lines 119–124: Break into three sentences for clarity:*

*"The paper then provides guidance on protocols for the mandatory Diagnostics, Evaluation, and Characterization of Klima (DECK) experiments and the recommended Assessment Fast Track experiments (Section 3). It distinguishes between experiments with a stronger emphasis on assessment and service-oriented prediction and projection, and those aimed at process understanding through characterization and attribution. The paper concludes with a discussion of CMIP's evolving role in the research community."*

**We have broken up the sentence and added further section titles.**

*• Lines 180–182: Consider revising for improved clarity as follows: "State-of-the-art coupled carbon cycle–climate modeling lies at the intersection of climate science, ecosystems, hydrology, biogeochemistry, and socioeconomic systems. The future resilience of natural systems and human-modulated carbon sinks remains one of the key uncertainties in efforts toward climate stabilization and warming reversal."*

**So taken**

*• Lines 213–216: Would this be better:*

*"Forest dieback and demographic shifts, for example, depend heavily on drought risk and related thermal and hydrological stressors (Drijfhout et al., 2015). This makes the representation of climate–vegetation interactions critical for robust assessments of potential change, especially as resilience may already be declining in the Amazon (Boulton et al., 2022)."*

**Taken with modification.**

*• Lines 284–285: "recommend extend the simulation out to 300 years" → "recommend extending the simulation to 300 years"*

**Accepted**

*• Line 310: Suggest breaking into two sentences: "…(if applicable). In other words,…"*

**Accepted**

*• Lines 328–331: Consider revising to:*

*"As background, modeling centers are advised to improve the historical $CO_2$ trend in their esm-hist simulations, addressing biases observed in the CMIP6 ensemble, which ranged from –15 to +20 ppm by 2014 (Gier et al., 2020). The causes of these biases and strategies for reconciling model output with observations have been the focus of extensive recent research (e.g., Hajima et al., 2025)."*

**The sentence has been split and revised.**

*• Lines 544–546: Consider revising to:*

*"Thematic diagnostic groups and sustained-mode initiatives are also being established, with teams focusing on the CMIP carbon footprint, controlled vocabularies, and quality control/assurance."*

**So taken.**